



# Toward a conceptual framework of hyporheic exchange across spatial scales

Chiara Magliozzi[1], Robert Grabowski[1], Aaron I. Packman[2], and Stefan Krause[3]

[1]Cranfield Water Science Institute, Cranfield University, Cranfield, MK43 0AL, UK
[2]Department of Civil and Environmental Engineering, Northwestern University, Evanston, Illinois, USA
[3]School of Geography, Earth and Environmental Sciences, University of Birmingham, Edgbaston, Birmingham, B15 2TT, UK

**Correspondence:** Robert Grabowski (r.c.grabowski@cranfield.ac.uk)

**Abstract.** Rivers are not isolated systems but interact continuously with groundwater from their confined headwaters to their wide lowland floodplains. In the last few decades, research on the hyporheic zone (HZ) has increased appreciation of the hydrological importance and ecological significance of connected river and groundwater systems. While recent studies have investigated hydrological, biogeochemical and ecohydrological processes in the HZ at bedform and reach scales, a compre-

hensive understanding of process-based interactions between factors operating at different spatial and temporal scales driving hyporheic exchange flows (HEF) at large and reach scale is still missing. Therefore, this review summarizes the factors and processes at catchment, valley and reach scales that interact and control spatial and temporal variations in hyporheic exchange flows. By using a multi-scale perspective, this review connects field observations and modelling studies to identify process driving patterns and dynamics of HEF. Finally, the influence of process interactions over multiple spatial scales is illustrated in

a case study, supported by new GIS analyses, which highlights the importance of valley scale factors to the expression of HEF at the reach scale. This conceptual framework will aid the development of approaches to interpret hyporheic exchange across scales, infer scaling relationships, and inform catchment management decisions.

## 1 Introduction

Hyporheic zones (HZ) are unique components of river systems that underpin fundamental stream ecosystem functions (Ward,

2016; Harvey and Gooseff, 2015; Boano et al., 2014; Merill and Tonjes, 2014; Krause et al., 2011a; Boulton et al., 1998; Brunke and Gonser, 1997; Orghidan, 1959). At the interface between rivers and aquifers, hyporheic zones are the expression of vertical and lateral connection of rivers with floodplains and the underlying aquifers, and are defined by the interchange of surface and ground waters through hyporheic exchange flows (HEF) (Lautz et al., 2010; Cardenas and Wilson, 2007; Gooseff et al., 2007; Boano et al., 2006; Wondzell, 2006; Wörman et al., 2006; Malard et al., 2002; Elliott and Brooks, 1997). HEF plays

a significant role in biogeochemical cycling (e.g., carbon and nutrient availability and transformation), ecological food webs, and habitat for diverse organisms (Ward, 2016; Merill and Tonjes, 2014; Krause et al., 2011a; Boulton et al., 1998; Brunke and Gonser, 1997). HEF is driven by potential and kinetic energy gradients near the streambed that change hydraulic head and force surface water to flow into, through and out of the bed (Boano et al., 2014; Cardenas et al., 2004; Elliott and Brooks, 1997). Both hydrostatic, i.e. sum of elevation head , and hydrodynamic forces, i.e. sum of velocity head, of the hydraulic head




contribute to HEF variations within rivers and floodplains (Boano et al., 2014; Cardenas et al., 2004; Packman and Brooks, 2001; Elliott and Brooks, 1997). Turbolence and (i.e. gravel bed substrate) and biological processes (i.e. bioturbation) also can drive HEF but less studied in steams and rivers (Boano et al., 2014). The hierarchical and heterogeneous nature of river and floadplain systems creates complex spatial and temporal patterns of exchange flows (Cardenas, 2008; Wörman et al., 2007).

There are strong gradients of structure and flow conditions formed by the drainage network that result from: (i) the temporal and spatial scales of the stream system from upstream to downstream, vertically and laterally (i.e. flood spates, overbank flows, etc.; (Minshall et al., 1985; Newbold et al., 1982, 1981), and (ii) complex geomorphological structures (armoring, bedforms, bars and other lateral variability within channels, braiding, meanders, floodplain deposits etc. . . ). Therefore, understanding and predicting HEF dynamics requires a consideration of the hydrological, topographical, hydrogeological, anthropogenic and

ecological processes operating across a spectrum of spatial and temporal scales (Ward, 2016). Previous work has identified multi-scale HEF processes, but has focused primarily on individual processes and controls within river corridors (Ward, 2016; Harvey and Gooseff, 2015; Boano et al., 2014). Existing information has not been synthesized to assess the multiple factors and characteristics that control HEF at catchment scales across geographic regions (Table S1,S2,S3 Supplementary Material). Similarly, earlier reviews have furthered our understanding of the ecological and functional significance of HZ (Krause et al.,

2011a; Boulton et al., 1998; Brunke and Gonser, 1997), the range of mechanisms and biogeochemical implications that influence HEF (Boano et al., 2014; Merill and Tonjes, 2014; Dent et al., 2001), and the challenges and perspectives to support interdisciplinary river research (Datry et al., 2017; Ward, 2016; Harvey and Gooseff, 2015). Despite this intensive investigation of HEF processes, there has been little investigation of hyporheic processes at catchment scale rather than at individual geomorphic units (Ward, 2016; Harvey and Gooseff, 2015; Krause et al., 2011a). Recently, Ward (2016) recognized that hyporheic

science is still facing the challenge of enabling cross-site comparisons of findings. One of the reasons is the absence of conceptual frameworks to translate patterns of hyporheic flows across scales, enable multi-scale assessment of process controls, and enable identification of common variables. Therefore, this paper reviews the state of knowledge of HEF with respect to two primary topics. With respect to the drivers of HEF, Sections 3-7 discuss five main drivers, hydrological, topographical, hydrogeological, ecological and anthropogenic, and how spatial and temporal variability in these drivers controls HEF. In the

context of multiscale interactions, Section 8 discusses how these drivers interact to create spatial and temporal heterogeneity in HEF direction and magnitude. Both sections highlight knowledge gaps that are important in terms of fundamental understanding and management of hyporheic zones. The review follows a hierarchical spatial approach, from reaches to catchment, and provides a structure upon which to explore the individual and interaction effects of factors on HEF and to upscale and downscale across spatially and temporally variable hyporheic processes (Fig.1, 2).

## 2   Concepts and terminology

The term "hyporheic zone" has been defined variously in the literature, and some confusion still exists within the wider research community about the extent and nature of the HZ. To help facilitate the integration and presentation of results from a large number of studies spanning a range of disciplines in this review, we will use a simplified and standardized terminology for





the HZ and hyporheic exchange flows (HEF). Herein, we follow the "flexible" definition of HZ, as reported by Ward (2016): "…saturated subsurface including flow paths that originate from and return to surface water where interactions occur within a temporal scale relevant to the process of interest, and processes of interest occur continuously from the stream–subsurface interface to the hyporheic–groundwater continuum." In terms of hyporheic exchange, we recognize that a continuum of hy-

porheic flow paths is associated with different hydrologic residence times (Boano et al., 2014; Cardenas, 2008; Wörman et al., 2007). In the context of multi-scale exchange, HEF is related to large-scale groundwater surface-water exchange (GSE), but the terms are not synonymous (Ward, 2016). HEF is an interchange between surface and subsurface waters occurring in short time scales (i.e. minutes to weeks), whereas GSE flows occur at a much larger scale and takes long times to return to the stream (i.e. months to millennia) (Toth, 1980). At the scale of HEF, GSE can be considered as unidirectional exchange (i.e. losing, river

recharges the aquifer, and gaining flow conditions, the river is fed by the aquifer). HEF and GSE can act in opposite directions (Stonedahl et al., 2012; Sawyer et al., 2009; Cardenas and Wilson, 2006). For example, a reach under losing condition due to groundwater (GW) recharge can have superimposed HEF occurring simultaneously (Fox et al., 2014; Stonedahl et al., 2012). Consequently, this review considers large-scale GSE in addition to HEF. We also refer to hyporheic "extent" when the HZ expands or contracts in the horizontal ("lateral extent") or vertical ("vertical extent") directions, respectively. Finally, we use

the term "bank storage exchange" for the case where lateral HEF between the river and floodplain is induced by the rise and fall of river water levels (Cranswick and Cook, 2015; Pinder and Sauer, 1971). Vegetation (i.e. vegetation density, riparian and in-channel vegetation) is considered in this review as the main ecological factor that influences HEF (Heppell et al., 2009; Corenblit et al., 2007). Although not reported in this paper, we acknowledge that other ecological factors such as hyporheic freshwater invertebrates and biofilm have a major role in interacting with HEF (Peralta-Maraver et al., 2018).

## 3 Hydrological drivers

Hydrological drivers influence HEF by changing surface- and ground-water flow regimes and distributions of hydraulic head. In this section, we provide a summary of how groundwater and river level fluctuations control the spatial and temporal distribution of hydraulic heads to affect HZ and HEF paths at reach (Section 3.1), valley and catchment scale (Section 3.2).

### 3.1 Groundwater and stream discharge at reach scale

HEF responds systematically to changes in hydrological conditions at reach scale. Together, river flow regime and event based fluctuations of groundwater levels control reach-scale hyporheic exchange by changing the distributions of hydraulic head (Boano et al., 2014). Several studies report that seasonal (i.e., spring-summer and summer-fall transition) and event-based changes in the gradient between river water and groundwater levels cause HZ to expand or contract (Malzone et al., 2016, 2015). In both losing and gaining flow conditions, the volume of the hyporheic zone contracts under a relatively small flux,

while hyporheic residence times decrease moderately (Fox et al., 2016). In particular, during gaining conditions, steep stream-ward hydrologic gradients limit the extent of the HZ (Fox et al., 2014; Wondzell and Gooseff, 2013; Cardenas, 2009; Cardenas and Wilson, 2007; Malcolm et al., 2005; Storey et al., 2003; Wroblicky et al., 1998; Harvey and Bencala, 1993a). Conversely,





the extent of the HZ and the hyporheic residence time increase during floods (Drummond et al., 2017; Zimmer and Lautz, 2014; Swanson and Cardenas, 2010; Wondzell et al., 2010; Poole et al., 2006). This enlargement is caused by the increases in stream stage and velocity, that in turn increase the exchange rate during the flood and drive water farther from the channel (Bhaskar et al., 2012; Malcolm et al., 2004). Conversely, inconsistent patterns of HZ have been observed in response to changes

to stream discharge (Ward et al., 2013; Wondzell, 2006). In mountainous streams, HZ was found to expand in small streams at lower base flow discharge (Q $< 0.01 \ m^3 \ s^{-1}$) compared to higher-discharge streams (Wondzell, 2011). This behavior has been interpreted to result from increasing hydrostatic head gradients associated with flow around channel morphological elements at low flow, such as development of lateral channels and flow around bars (Wondzell, 2006). Consistently with the above findings, HEF paths do not respond uniformly to stream discharge and groundwater flow at reach scale. Groundwater discharge reduces

HEF flux and flow path residence time and length, while stream discharge alone does not affect significantly HEF length and residence time (Schmadel et al., 2017; Gomez-Velez et al., 2015; Boano et al., 2008; Cardenas and Wilson, 2007). In spatially heterogeneous reach morphology, these responses are exacerbated by the presence of reach morphological features ((Dudley-Southern and Binley, 2015; Zimmer and Lautz, 2014); Fig. 4 in (Schmadel et al., 2017)). Schmadel et al. (2017) observed that flow paths generated by large hydraulic gradients (i.e. bedforms) are less sensitive to changes in hydrological conditions

than those generated by the larger context of the valley gradient (Schmadel et al., 2017). Such complex interactions between groundwater and river regimes generally makes it difficult to identify the dominant drivers of HEF without considering multiple spatial scales. To develop frameworks with improved spatio-temporal resolution of HEF, comprehensive understanding of the valley hydrological condition is required.

### 3.2 Groundwater and stream discharge in a large scale

Interactions at the reach scale between the factors described in Section 3.1 often results in heterogeneous responses of HEF that require the consideration of processes at a larger. HEF and residence time in river reaches are affected by the relationship between hillslope structure and hillslope water table (Hoagland et al., 2017; Torres et al., 2015; Godsey, 2014; Jencso et al., 2010). To date, model simulations have showed that diel fluctuations of hillslope water tables affect both the length and the residence time of HEF. These fluctuations, which occur due to the temporal lag between stream and aquifer responses,

produce a wide range of hydraulic gradients (Wondzell et al., 2010, 2007) and affect HEF by several orders of magnitude. Longer hyporheic flow paths result in locations with larger hydraulic conductivity, large stream amplitude and large hillslope amplitude with respect to the stream (Schmadel et al., 2017, 2016). Given the diverse geomorphology of river valleys and the seasonal responses of hillslope water table fluctuations to large scale controls (e.g., precipitation), the relationship between dynamic hydrological valley conditions and HEF remains an area of active research (Schmadel et al., 2017; Bergstrom et al.,

2016; Schmadel et al., 2016; Nippgen et al., 2015; Wondzell and Gooseff, 2013; Jencso et al., 2009). Recent studies have started to consider precipitation inputs to the catchment to enable cross-catchment comparisons of HEF (Hoagland et al., 2017; Jasechko et al., 2016). The drivers discussed in Section 3.1 and 3.2 vary within and among catchments depending on catchment topography, geology and finally geography (Hoagland et al., 2017; Jasechko et al., 2016). For example, steep, headwater catchments respond rapidly to rainfall because of their small storage capacity (Penna et al., 2016; Gomi et al.,



2002; Woods et al., 1995). Rainfall is strongly correlated with seasonal groundwater fluctuations in catchments dominated by transmissive soils (Bachmair and Weiler, 2012). Conversely, in headwater catchment with low permeability soils, rainfall is only a secondary control, after topography, on the response time of groundwater levels (Rinderer et al., 2016). On the other hand, lowland catchments have usually slower response to rainfall (days to weeks), although heavy precipitation events can

cause local flooding (Monincx, 2006). Finally, the relationship between groundwater, stream discharge and HEF is dynamic in nature, depending on the cross-scale interaction of hydrological gradients. Thus, HEF findings at reach scale may not be representative when major changes, e.g., seasonal variations, occur in valley- or catchment-scale characteristics.

## 4   Topographical drivers

Topography is one of the primary drivers of spatial HEF variability. From bedforms to catchments, topographic gradients cause

nested hyporheic flow paths (Cardenas, 2008; Wörman et al., 2007). In order to understand how HEF varies spatially within the catchment and how these variations in turn affect temporal variations, we will discuss HEF at scales within channel topography (individual bedforms and bedforms sequences: Sections 4.2 and 4.3), within a valley hydrological (bedforms in valley context: Section 4.4) and geomorphological context (valley confinement: Section 4.5) and then within the catchment (Section 4.7).

### 4.1   In-channel bedforms

Over the last decades, a range of studies have demonstrated that hydrodynamic and hydrostatic forces generated by in channel bedforms have large effects on the variability of HEF from cm to m scale. In this section, we provide a concise summary of the main effects on HEF by single bedforms (i.e. steps, riffles and bars) and bedforms sequences (i.e. step-pool, pool-riffle). We considered bedforms that induce hydrodynamically-driven HEF, i.e. ripples and dunes (Section 4.2), and larger topographic features, i.e. steps, riffles and bars, that contribute to hydrostatically-driven HEF (Section 4.3) (Boano et al., 2014).

### 4.2   HEF generation by an in-channel bedforms

Head pressure gradients created by the channel bedforms drive advective pore water flow into, though, and out of the bed (Elliott and Brooks, 1997). Most of the current knowledge of hyporheic fluxes and their spatio-temporal variability in submerged bedforms has been obtained from simulations and laboratory experiments, owing to the difficulties in making high-resolution subsurface measurements within stream channels (Boano et al., 2014; Fox et al., 2014; Irvine et al., 2014; Trauth et al.,

2014; Stonedahl et al., 2013; Janssen et al., 2012; Cardenas and Wilson, 2007; Tonina and Buffington, 2007; Glaser et al., 2004; Elliott and Brooks, 1997). Bedforms develop characteristic shapes due to the interplay of stream flow and bed sediment transport. Dunes and ripples are characterized by a smooth water surface profile (Packman et al., 2004) implying that the spatial variation of water surface topography is minimized and the pressure profile strongly depends on dynamic pressures (Marion et al., 2002; Elliott and Brooks, 1997). In the case of hydrostatical-driven HEF, the flow is a function of the head gradient,

the size and the hydraulic conductivity around the bedform (Hester et al., 2008; Gooseff et al., 2006). High channel slope will normally result into deeper HEF and higher HZ depth (Hester et al., 2008; Gooseff et al., 2006). Riffle-scale exchange, for





example, is possible only when high permeability materials surround the stream channel. Hyporheic flow structure is controlled by spatial relationship of bedforms to high- and low-permeability regions of the streambed (Stonedahl et al., 2018; Pryshlak et al., 2015; Sawyer and Cardenas, 2009; Packman et al., 2004; Salehin et al., 2004). Water upwells where permeability or depth of gravel decreases in the direction of streamflow and where the longitudinal bed profile is concave (Buffington and

Tonina, 2009; Elliott and Brooks, 1997; Harvey and Bencala, 1993a). Water downwells where permeability or depth of gravel increases, in the direction of streamflow, or where the longitudinal bed profile is convex (Buffington and Tonina, 2009; Elliott and Brooks, 1997; Harvey and Bencala, 1993a). Modelling studies have showed that flow paths and exchange rate vary in the alluvium around riffles across seasons and with the extent of groundwater discharge (Stonedahl et al., 2018, 2012; Storey et al., 2003). Gravel bars are also functionally equivalent to riffle bedforms for HEF; the hydrologic retention in gravel bars

is strongly influenced by bar structure and stream water levels (Trauth et al., 2015; Tonina and Buffington, 2007; Marzadri et al., 2010; Boulton et al., 1998). Unlike fully submerged features, recent findings by Trauth et al. (2015) suggest that HEF in partially submerged gravel bars decreases with increasing stream discharge as the hydraulic head gradients across the bedform decrease, leading to long residence times under low flow conditions. In conclusion, an in-channel bedform can have significant effects on its own on HEF and on its residence times distributions. More complex interactions are expected to occur across the

spectrum of topographic features (Stonedahl et al., 2010).

## 4.3 In-channel bedform sequences

The complexity of nested hyporheic flows will increase with the number and diversity of bedforms in the channel. Local-scale variation of bedforms size will drive longitudinal patterns of upwelling and downwelling, along with multiscale distributions of HEF at reach scale (Stonedahl et al., 2015, 2013, 2010; Gooseff et al., 2006). Step-pool morphology behaves differently

than pool-riffle and dune-like bedforms (Hassan et al., 2015; Marzadri et al., 2010; Tonina and Buffington, 2007; Storey et al., 2003). HEF will develop around a pool-riffle sequence only where hydraulic gradients toward the stream from the sides and beneath are less than or near than the longitudinal hydraulic gradient between the upstream and downstream ends of the riffle (Storey et al., 2003). In gravel bed pool-riffle sequences, significant hydrostatic forces across the channel, high permeability of sediment and low submergence time generate substantial large-scale hyporheic flow (Tonina and Buffington,

2011, 2007; Buffington and Tonina, 2009; Wondzell and Swanson, 1996). A detailed case study on a upland, gravel-bed river with a riffle-pool bedform sequence showed that, although the expected pattern of downwelling and upwelling conditions were generally observed along the bedform sequence, seasonal variations in hyporheic fluxes occurred because of asynchronous local ground water recharge relative to flow regime (Gariglio et al., 2013). At riffle-pool scale, this is consistent with previous studies reporting seasonal variations in hyporheic temperature dynamics, with stream topography, sediment stratification, and

groundwater interaction all affecting local upwelling and downwelling in riffle-pool systems (Krause et al., 2013; Hannah et al., 2009). Dune-ripple complexes are less influenced by hydrostatic forces than riffle-pool sequences (Tonina and Buffington, 2011); gradients are much lower than for riffle/pool and step/pool sequences and little affected byspatial and temporal changes in water surface elevation. Simulations have also shown that dunes contribute more than meanders and bars to reach-scale HEF (Stonedahl et al., 2013). Further, the volume of water exchanged and the hyporheic residence time across bedforms is not



linearly additive (Stonedahl et al., 2013). Instead, hyporheic exchange is maximized when one topographic feature dominates (Stonedahl et al., 2013). In lowland rivers, the lower slope, finer sediments and more constant flows favor the development of dune-ripple sequences (Elliott and Brooks, 1997; Marion et al., 2002) characterized by high relative submergence and smooth water surface profiles (Packman et al., 2004). Under these conditions, the spatial variation of water surface topography is

minimized and HEF is induced primarily by dynamic pressure variations. These findings suggest that in-channel bedforms often control HEF, although these local exchange flows are still strongly modulated by stream and groundwater dynamics at reach and valley scale.

## 4.4 Alteration of in-channel bedform induced HEF by valley hydrology

The patterns of HEF generated by individual bedforms and bedform sequences are altered by the hydrodynamic conditions

of the valley. Longitudinal valley gradients create hydrostatic head gradients that influence water moving cross and down valley and thus HEF (Harvey and Bencala, 1993a). Schmadel et al. (2016) suggested that valley slope primarily controls the timing of HEF while cross-valley slope and down-valley slope determine net gaining or losing conditions. When bedforms are analyzed with respect to channel gradient, it can be seen that gentle slopes of lowland rivers generate slower currents with deeper flows, lower relative roughness, and less valley confinement, resulting in less bedform-induced exchange (Tonina

and Buffington, 2007) (Fig. 3). For example, dune-ripple streams that occur in lowland rivers, typically exhibit less spatial and temporal variability in water surface elevation than riffle-pool streams (Tonina and Buffington, 2011). In higher-gradient valleys, the flow is predominantly down-valley and spatial variations of hydraulic gradients are paired with changes in cross-sectional areas of the valley and with the hydrodynamic head gradients generated by in-channel bedforms to induce water downwelling into the HZ (Wondzell, 2012; Cardenas et al., 2004). In this setting, hydrogeological properties can have a

major role in controlling valley hydrologic exchange: Ward et al. (2012) and Anderson et al. (2005) observed that in steep and constrained sections of his study area, the HEF in step-pool sequences is limited by the underlying bedrock rather than by hydraulic gradients. In conclusion, both positive and negative relationships between hyporheic zone extent and down- and cross- valley gradients have been reported in literature, suggesting that detailed resolution of hydraulic gradients and knowledge about the valley setting are necessary to understand controls on HEF (Ward et al., 2012).

## 4.5 Valley confinement

The extent of valley confinement indicates different process domains and determines the capacity of the river to adjust in planform (Table 1). Several studies have linked HEF to valley confinement, and showed that HZ depth is restricted, HEF is reduced, and hyporheic residence time is decreased in highly confined valleys (Buffington and Tonina, 2009; Wright et al., 2005; D'angelo et al., 1993; Stanford and Ward, 1993)(Table 1). While GSE and HEF are both limited in confined valleys,

bedrock fractures and fissures may allow some hyporheic exchange, depending on their degree of connectivity with the aquifer (i.e. bedrock and colluvial channels in straight and sinuous planforms) (Gurnell et al., 2016; Graham et al., 2010; Freer et al., 2002; McDonnell et al., 1997, 1996). Certainly, the coupling of small changes in water table elevation and bedrock topography can have a large impact on the hyporheic flows (Oxtobee and Novakowski, 2002). For example, HEF transport is expected





to be more uniform when the water table is continuous on the bedrock than when the water table falls and interacts directly with bedrock topography (Ward et al., 2012). Bedrock outcrops at valley margins can have opposing impacts on HEF. On one hand, they can limit the infiltration of the stream water into the subsurface and restrict the hyporheic zone (Kasahara and Wondzell, 2003). Indeed, bedrock outcrops can constrain valleys where steep positive vertical hydraulic gradients results from

discontinuities of superficial deposits permeability and shallow bedrock (Ibrahim et al., 2010). In this case the HEF can be limited to superficial layers of the riverbed. On the other hand, the irregularities of bedrock projections favor changes in the alluvium volume (Buffington and Tonina, 2009), thus driving stronger hyporheic exchange from the subsurface to the stream and preventing deeper GSE. In fact, the interchange between bedrock and alluvial valleys favors HEF, because of increased downwelling and upwelling where a thin layer of alluvial deposits overlies shallow bedrock (Ward et al., 2012; Wondzell,

2012). Conversely, in unconfined valleys, floodplain sediments typically represent a mosaic of coarse and fine sediments that originate from hillslopes, bed material (i.e. bedload) and suspended sediment deposited during overbank flooding, within the context of channel adjustment over time (e.g., migration and avulsion) (Nanson and Croke, 1992)(Table 1). Tonina and Buffington (2009) classified channel types by examining how bedforms generate hydrodynamic pressure variations and drive hyporheic exchange (Fig.3). Generally, unconfined channels have smaller vertical hydraulic gradients and discharges than

confined channels, caused by the lower channel gradients and by the heterogeneity of sedimentary deposits (Ibrahim et al., 2010). We synthetize available information on underlying geology, in-channel sediment, valley confinement at valley and reach scales in Table 2, where for different channel planforms, geomorphic units and floodplain characteristics potential HEF response is indicated. In conclusion, empirical and modelling studies not only suggest the dominance of hydrologic exchange flows by small geomorphic features but also that lateral exchanges of water affect movement of material and energy between

rivers and floodplains.

## 4.6 Channel planform

As valley confinement (Section 4.5), channel planform is an indicator of lateral HEF interactions with floodplains. Sinuosity is often used as a measure of channel complexity and has been found to be directly correlated with lateral hyporheic exchange in meander bends, and in the parafluvial zone beneath the streambanks (Kiel and Bayani Cardenas, 2014; Cardenas, 2008; Boano et al., 2006; Wroblicky et al., 1998; Holmes et al., 1996). Sinuosity establishes pressure gradients across meanders that induce HEF (Boano et al., 2008, 2006) and influences the amount of water exchanged within a river segment (Han and Endreny, 2013; Gomez et al., 2012; Cardenas, 2009; Brunke and Gonser, 1997). High sinuosity rivers (e.g., multi-thread or single/sinuous meandering) are less prone to a reduction of the hyporheic area with depth, and maintain the HZ under both losing and gaining conditions (Cardenas, 2009) (Table 1). Meander planimetry drives hyporheic flows and influences hyporheic

residence times by creating differences in the elevation head of surface water around a meander bend, with spatial and temporal variations as meanders evolve (Stonedahl et al., 2013; Boano et al., 2008; Revelli et al., 2008; Boano et al., 2006). Naturally forced by the longitudinal head gradient, the hyporheic exchange flows through the meander neck as river water infiltrates into the hyporheic zone at the upstream half of the meander and returns to the river along its downstream half (Kiel and Bayani Cardenas, 2014; Boano et al., 2006; Cardenas et al., 2004). This pattern becomes more complex with the inclusion of floodplain



sediment and channel geomorphic features. Lateral hyporheic residence time is short in areas with coarse floodplain sediments and high sediment hydraulic conductivity, and increases in meanders with fine-textured sediments (Boano et al., 2006). In multi-thread planforms, simulations have identified the importance of hyporheic flow paths beyond the active channels toward secondary channels and across the floodplain ((Kasahara and Wondzell, 2003), Table 1). Along laterally unconfined valleys, meander creation, extension and cutoff allow significant river adjustment and river-floodplain interactions, causing both in-stream and off-channel geomorphic features to drive lateral hyporheic exchange (Boano et al., 2006). In conclusion, studies of valley setting, confinement and sinuosity suggest that valley topography provides important clues about disconnection within catchments and can be potentially used as a quantitative and quantitative predictor of HEF. As demonstrated by the above studies, the source of spatial complexity of HEF is not only the result of single geomorphic structures but of the topographical structure of the valley and of the whole catchment.

## 4.7 HEF in the catchment topography context

Studies have suggested that catchments with larger surface areas have greater hyporheic exchange fluxes (Bergstrom et al., 2016; Laenen and Bencala, 2001; Harvey and Wagner, 2000). Greater variation in water stage correlates on average to greater hyporheic fluxes, but few direct observations are available to support or refute this assumption. The catchment topographic slope defines the direction of flow by creating discontinuities and localized groundwater flow paths (Jencso and McGlynn, 2011; Jencso et al., 2009; Wörman et al., 2006; Winter, 1998). Emerging upscaling models have started incorporating the information of the catchment area, channel network structure, and head variations of surface topography. These models include i) the first order control of water inputs and groundwater head distribution (Caruso et al., 2016; Jencso and McGlynn, 2011; Laudon et al., 2007), (ii) indications of subsurface flow (Caruso et al., 2016; Jencso and McGlynn, 2011; Jencso et al., 2009; Wörman et al., 2006); and (iii) discretizing the catchment into sub-catchments and identifying topographically contributing recharge and discharge areas (Wörman et al., 2007, 2006). These studies indicate that linking topographic complexity to HEF is likely to be an important forefront area of research. Patterns of upwelling and downwelling within reaches were observed to occur where stream profile is concave and convex, respectively and used to predict patterns of HEF in high-gradient headwater mountain streams (Anderson et al., 2005). While upwelling zones do not show a significant trend with increasing catchment area, the length of downwelling zones increases with stream size, spacing of channel slope and decrease of water surface concavity (Anderson et al., 2005). These findings encourage interdisciplinary efforts to provide supporting evidence that link HEF across the continuum of headwater, mid-order and lowland streams as result of systematic changes in hydrogeomorphological characteristics along the stream network.

## 5 Hydrogeological drivers

Geology affects both the distribution of groundwater in aquifers and HEF flows. In this section hydrogeological effects on HEF are summarized into: i) channel sediment impacts on bedform-induced HEF (Section 5.1), ii) floodplain sediment im-





pacts on GSE between the valley aquifer and the channel (Section 5.2), and iii) bedrock and aquifer type impacts on valley geomorphology (Section 5.3).

## 5.1 Channel sediment and bedform-induced HEF

Sedimentological properties strongly control HEF at reach scale. Water flowing through the river bed is affected by sediment grain size, sediment heterogeneity, and depth, promoting spatially diverse hyporheic exchange (Packman et al., 2004). Given the direct coupling of stream and pore water flow, exchange is generally greatly enhanced in coarser sediments (Packman et al., 2004). As mentioned in Section 1, high velocity gradients and turbulence generated at the surface of coarse sediment beds can also increase diffusion processes which can produce considerable exchange even when the bed surface is flat and no flows are induced by bed topography (Marion et al., 2008; Packman et al., 2004). The presence of high hydraulic conductivity layers in the streambed increases dispersive mixing between hyporheic water and groundwater (Hester et al., 2013) and creates preferential HEF, either short or long paths, by controlling the ability of the sediment to support advective pumping (Pryshlak et al., 2015; Cardenas, 2009, 2008; Salehin et al., 2004). Dye injections have shown that hyporheic flow patterns are controlled by the spatial relationship of high and low permeability regions of the streambed, resulting in faster near-surface transport and shallower penetration and a shorter mean residence time (Salehin et al., 2004). Further, longer hyporheic flow paths are generated in streams having greater connectivity of sediment strata (Pryshlak et al., 2015) despite that in coarser bed material, fine sediments accumulate and clog pores (Hartwig and Borchardt, 2015; Bardini et al., 2012; Brunke and Gonser, 1997). To date, few studies have addressed the effect of sediment heterogeneity on HEF variability at scales larger than the bedform, although recent works have showed strong impact of sand and gravel deposits on HEF at the reach scale (Zhou et al., 2014) and identified sediment heterogeneity as one of the main drivers of lateral connectivity as well (Pryshlak et al., 2015). In river segments dominated by gravel beds, such as in confined high-energy braided rivers, the hydraulic conductivity is generally high but also highly variable because it depends on the sorting of sediments in the floodplain and on the amount of silt and clay present (Table 1 and Fig. 3). Highly permeable riverbed sediments allow surface water to penetrate easily into the HZ, causing vertical hydraulic gradients (VHG) to change strongly with local sediment permeability (Packman and MacKay, 2003; Wroblicky et al., 1998; Vaux, 1968).

## 5.2 Hydrogeology in river and floodplain type

Channel planforms respond not only to changes in regional physiography and hydrology (Section 4.6) but also to sediment loads (Table 1) (Gurnell et al., 2016; Nanson and Croke, 1992). Differences in particle sizes in river planforms result in fact, from longitudinal, lateral, spatio-temporal variation of river flows and sediment supply (Bridge, 2009; Baldwin and Mitchell, 2000). Sediment permeability allows varying hyporheic residence time responses accordingly to finer or coarser deposits (Hester et al., 2016; Pryshlak et al., 2015; Azinheira et al., 2014; Brunke and Gonser, 1997) (Fig. 3). Braided channels (Section 4.6) can occur across a range of valley slopes depending on the grain size of the bed material in transport, and present either a pool-riffle morphology or a bar-riffle morphology (Gurnell et al., 2016). HEF tends to be very dynamic and spatially varying; steep head gradients between channels create cross-valley head gradients that control the location and direction of flow paths through





the HZ (Fig. 3, Section 4.6) (Malard et al., 2002; Ward and Stanford, 1995). This transverse exchange evolves with migration and river sediment transports processes (Stonedahl et al., 2010; Boano et al., 2006; Kasahara and Wondzell, 2003). In sinuous, medium energy meandering floodplains, HEF is also usually driven by variations in head gradients (advection processes), which are greater than diffusive transport by two or more orders of magnitude (Elliott and Brooks, 1997; Larkin and Sharp, 1992). This type of floodplain typically presents vertically accreted fine sediments (silt and clay). These local low-permeability units and thick sequences of unconsolidated deposits become more compact and less permeable with depth (Winter, 1998) thus they are characterized by localized groundwater flows and restricted HEF (Angermann et al., 2012; Krause et al., 2012; Stonedahl et al., 2012). In lowland settings with abundant fine sediment load, reduction of groundwater up-welling due to low sediment conductivity layers causes surface water to downwell and induces horizontal hyporheic flow into shallow streambed sediments above low conductivity strata (Angermann et al., 2012; Stonedahl et al., 2012). Spatial variations in the thickness of fluvial-alluvial deposits increased local gradients around clay lenses, therefore creating locally confined conditions (Ellis et al., 2007). All of these studies indicate that the thickness of superficial deposits controls the extent and rate of hyporheic exchange (Tonina and Buffington, 2011; Buffington and Tonina, 2009; Anderson et al., 2005).

### 5.3 Hydrogeology in the catchment: bedrock and aquifer type

From reach to regional scale, geology affects the distribution of groundwater in aquifers and the spatial variability of GSE and HEF via the aquifer geometry and hydrogeological properties. Lithologic types and structure, weathering history of bedrock and types of aquifers, impact HEF by altering the distribution of hydraulic conductivities (Fox et al., 2014; Gomez-Velez and Harvey, 2014; Angermann et al., 2012; Krause et al., 2011b; Hiscock, 2007; Woessner, 2000; Morrice et al., 1997; Winter, 1998). Bedrock exerts vertical and lateral constraints on river forms and processes, by controlling the interaction of GSE and HEF subsurface flows and defining valley confinement (Section 4.5). Different relationships appear depending on whether the structure is consolidated or semi-consolidated, and on the primary and secondary porosity of rock deposits including limestone, dolomite, shale, siltstone, sandstone, and conglomerate (e.g., pores and fractures) (Binet et al., 2017; Hoagland et al., 2017; Jencso et al., 2010; Sear et al., 1999). For example, hyporheic studies in chalk catchments have shown the importance of groundwater in supporting surface-subsurface exchange at catchment (Lapworth et al., 2009; Grapes et al., 2005), valley and reaches scales (Griffiths et al., 2006; Grapes et al., 2005), although vertical hyporheic exchange in these systems is often restricted by local low-conductivity superficial deposits (Allen et al., 2010; Pretty et al., 2006) (Section 5.1). In addition to the characteristics of the bedrock, the degree of confinement of the aquifer due to impermeable layers would prevent or limit GSE and HEF to local interactions (Gurnell et al., 2016). In confined aquifers, which are separated from the surface by aquitards with low hydraulic conductivities, GSE would likely be prevented (Winter, 1998). If the confinement is due to the presence of near-surface bedrock, HEF would also be prevented by the lack of highly porous alluvium and the low permeability of the bedrock (Buffington and Tonina, 2009; Kasahara and Wondzell, 2003). In confined bedrock, colluvial channels, and confined alluvial channels, GSE and HEF are limited by the local structure of the local sediment (e.g., coarse or fine particles) and the rock structure (e.g., continuous or discontinuous confinement) (Table 7.5 in Gurnell et al. (2016)). In unconfined aquifers, generally groundwater is easily conveyed in all directions leading to high opportunity for both vertical and lateral HEF exchange (Winter,



1998). However, in unconfined alluvial channels, GSE and HEF can be prevented or limited to local interactions depending on local sediment (e.g., coarse or fine particle size) (Table 7.5 in Gurnell et al. (2016)). In conclusion, HEF from reach to catchment scales is highly related to bedrock lithology and superficial sediment. The complexity of geological properties at the catchment scale results in spatio-temporal variations in HEF, in the channel and throughout the river network. A point upstream

in the catchment may exhibit HEF dynamics driven by entirely hydrogeological processes compared to the catchment outlet. These differences are especially heightened in catchment with mixed land use and anthropogenic pressures (e.g., dams) for which comprehensive understanding is required of the timescales of water and solute flux with different geologies.

## 6  Ecological drivers

Vegetation has long been known to exert a strong control on land surface hydrology by moderating streamflow and groundwater

recharge (Section 3.2). By altering hydrological processes on channel banks, floodplains and the wider catchment, vegetation-induced feedback on the temporal variability of HEF and likely increase the spatial heterogeneity of this ecological- hydrological relationship. This section describes in-channel (Section 6.1), bank and floodplain vegetation by focusing on two key ecological functions: riparian vegetation (Section 6.3) and large in-channel wood (Section 6.2).

### 6.1  In-channel vegetation

In-channel vegetation controls HEF directly through channel-scale flow resistance and indirectly through sediment and streambed permeability (Jones et al., 2008). A variety of herbs, shrubs and trees grow in stream channels, increase bed roughness and alter flow velocities. They produce a mosaic of hydrodynamic conditions with low flows in vegetation patches and high flows between patches (Corenblit et al., 2007). Vegetation also alters stage-discharge relationships that affect hyporheic flow, where higher water levels and faster in-channel flows are maintained in mid-summer (Heppell et al., 2009; Harvey et al., 2003). Jones

et al. (2008) demonstrated that in-channel vegetation restructures hyporheic flow patterns by creating temporally dynamic deviations of hydraulic gradients. Certainly, in-channel vegetation increases the friction factor (Harvey et al., 2003) and the create low flow areas that increase water residence time (Kjellin et al., 2007; Ensign and Doyle, 2005; Wörman and Kronnäs, 2005; Salehin et al., 2003). This aspect has been observed especially in streams with extensive vegetation where flow can decrease to nearly zero within dense vegetation stands (Ensign and Doyle, 2005; Salehin et al., 2003). Further, the reduction of flow

velocity within plant stands leads to increased sediment deposition and the development of plant-mediated sediments that are typically finer-grained than the bed material with more organic material and lower permeabilities (Corenblit et al., 2007), which also reduces HEF. In conclusion, both field and laboratory studies have suggested that vegetation shapes transient storage in streams channels, even though there are still difficulties in understanding the feedback of mixing due to vegetation and to induced HEF at reach scale. The role of vegetation on patterns of HEF at larger spatial scales is still unexplored. In particular,

bank vegetation needs to be considered in terms of hydrological connection between riparian vegetation and the stream (Duke et al., 2007) (Section 6.3).



## 6.2 In-channel wood

Within stream channels and valleys, wood deposits drive physical complexity of the river network by altering flow resistance, channel-floodplain connectivity, vertical and lateral accretion of floodplain (Davidson and Eaton, 2013; Wohl, 2013; Phillips, 2012; Jeffries et al., 2003; Mutz, 2000; Sear et al., 1999; Piégay and Gurnell, 1997). Wood affects channel hydraulics and induces deeper HEF by increasing the variability in vertical head and imposing greater hydraulic resistance (Lautz and Fanelli, 2008; Mutz et al., 2007; Mutz, 2000). Wood generally has a comparable effect to other in-channel structures (Section 7.1) and channel roughness elements (Section 4.3) by driving water into the subsurface, where it travels along short hyporheic flow paths (Boano et al., 2006; Lautz et al., 2006). The impact of wood on HEF varies with valley topographic gradient (lowland and upland), groundwater dynamics (gaining and losing) and sediment transport (Gregory et al., 2003; Jeffries et al., 2003). In lowland rivers, where flow velocity is slow and gradient low, wood induces less HEF and also has less effect on spatial patterns of HEF (Krause et al., 2014). Temporally, Wondzell (2006) observed that, although lowland streams are sensitive to changes in wood delivery, and wood decreases HEF at short time-scales, large-scale channel adjustments reverse the effect of natural wood removal over longer time-scales, causing higher HEF fluxes. Over the long term, wood removal results in longer mean hyporheic residence times, which impacts many hyporheic functions including temperature, nutrient retention, and oxygen concentrations (Sawyer and Cardenas, 2012; Stofleth et al., 2008). In upland rivers, wood typically creates steeper head gradients that drive hyporheic flow paths (Krause et al., 2014). Interactions between flow and wood also produce spatial heterogeneity in deposits of sediments and organic matter (Osei et al., 2015b, a; Sear et al., 2010; Latterell et al., 2006; Naiman et al., 2000; Gregory et al., 1991). Fines and organic-rich sediments are retained, eventually driving higher spatial heterogeneity in HEF (Section 5.2 and 7). However, Kasahara and Hill (2006) observed little impact of a large wood-constructed step on oxygen concentrations within the hyporheic zone, presumably due to siltation (Parker et al., 2017; Wohl et al., 2016; Menichino and Hester, 2014). At the valley scale, wood delivery depends on short- and long-term patterns of land use and geomorphology, often establishing floodplain geomorphology as the dominant control on wood storage in river systems (Benda and Bigelow, 2014). Indeed, one of the variables influencing wood transport and storage is valley geometry. Several studies have documented the importance of woody debris in shaping channel patterns and floodplain evolution in a variety of environments (Collins et al., 2012; Millington and Sear, 2007; Abbe and Montgomery, 2003; Jeffries et al., 2003; Collins et al., 2002; Piégay and Marston, 1998; Sear et al., 2010). However, relatively few studies have examined patterns of wood distribution relative to valley geometry or HEF responses to morphological changes induced by large wood at valley scale (Wohl and Cadol, 2011).

## 6.3 Riparian vegetation

At valley scale, riparian vegetation is well known to shape patterns of GSE by affecting riverbank filtration and altering water-table elevations via transpiration (Jones et al., 2008; Chen, 2007). Vertical and lateral hyporheic flow patterns are characterized by non-linear spatial variations with both vegetation composition (i.e., species) and water consumption (i.e., ET) (Larsen et al., 2014; Wondzell et al., 2010; Martinet et al., 2009). The ET from riparian vegetation can increase hyporheic fluxes by 1-2 orders of magnitude at time scales of weeks to months (Larsen et al., 2014). The effect of ET on HEF is especially significant in low-



energy environments, where ET drives mixing comparable to that of molecular diffusion and that varies at different times of the year (Bergstrom et al., 2016; Larsen et al., 2014; Iturbe and Porporato, 2004; Porporato et al., 2004). Conversely, in high-energy environments where turbulent mixing and bedform-induced pumping are very rapid (Section 4.4 and Fig. 3), the effect of ET will be lower. On the daily time scale, evapotranspiration changes groundwater gradients with riparian zone vegetation

creating the lowest water table in the afternoon, promoting surface water infiltration and hyporheic exchange (Wondzell et al., 2010; Loheide and Lundquist, 2009). Duke et al. (2007) observed a seasonal correlation between transpiration and stream flow with hyporheic gradients. During winter, the correlation is very strong and high water tables and hillslope vegetation lead to negative hyporheic gradients and to high hydraulic head at the bank surface. Conversely, in summer the stream channel has less surface flow and less active exchange within the HZ, and deep flow paths are very important in this period (Duke et al.,

2007). At valley scale, the effect of riparian vegetation has been observed to greatly influence energy inputs to the stream by controlling channel complexity, resulting in increased retention by increasing residence time and contact between stream water and hyporheic zone. This hydrological interaction has been studied in arid catchments (i.e. Sycamore Creek, a Sonoran Desert stream – (Schade et al., 2005, 2002)) where soils are often highly impermeable and the presence of riparian vegetation is dependent on stream flows and shallow groundwater tables (Schade et al., 2005, 2002; Stromberg et al., 1996). Most of these

studies have been performed in arid environments, and information on the effects of ET on HEF in humid environments is lacking. In conclusion, the direct and indirect effects of riparian vegetation on HEF at floodplain/catchment level are poorly studied relative to effects of morphology and groundwater recharge/discharge, although the studies mentioned above provide a foundation for evaluation of groundwater-dependent riparian vegetation on the HZ.

## 7   Anthropogenic drivers

Humans have extensively modified many river systems, and these changes impact the natural factors and processes that control HEF. Alterations to catchments, valleys, and river channels affect the hydrology (e.g., river stage fluctuations), hydraulics (e.g., altering vertical hydraulic gradients) and physiographic setting (e.g., geology, morphology). Effects of three main anthropogenic factors on HEF will be discussed: (i) river stage fluctuations due to in-channel structures and (ii) valley-spanning dams, and (iii) changes in sediment delivery and channel complexity due to land use and land management.

### 25   7.1   In-channel structures

Channel structures (e.g. weirs, log dams) that control change flow conditions by obstructing the flow and dissipating energy have positive and negative impacts on HEF (Daniluk et al., 2013; Hester et al., 2008; Lautz et al., 2006). Upstream of the control structure, a decrease in channel velocities and bedform size, combined with an increase in water depth and channel cross-sectional area are usually observed and associated with a reduction of turbulent hyporheic exchange in coarser sediments

(Blois et al., 2014; Boano et al., 2010; Jin et al., 2009) and advective HEF by ripples, dunes, and bars (D'angelo et al., 1993). Downstream of control structures, a decrease in sediment loads, scour, and turbulent fluxes in coarser sediment are usually observed (Hester et al., 2009). Weirs induce HEF upstream of the obstruction, flow beneath it, and upwelling on the downstream





side (Jin et al., 2009; Hester and Doyle, 2008). The effect of these structures is complicated and may vary under different flow conditions. Conservative tracer experiments at reach scale have showed that the cumulative effect of multiple weirs increased the cross-sectional area of the surface stream and of the transient storage zones behind weirs, while HEF decreased (Rana et al., 2017). As a consequence, multiple weirs reduce short and fast HEF while inducing long and slow-moving hydrostatically-

driven hyporheic flow paths (Rana et al., 2017). Hence, the evaluation of potential effects of channel-spanning structure on HEF requires rigorous analysis with respect to channel flow variation. The various effects of these measures are complicated and include disruption of downstream flux of sediment with critical consequences for the alluvial structure and on HEF at streambed or meander scale (Poole and Berman, 2001).

## 7.2 Dams

Large valley-spanning obstructions such as dams can affect HEF by ponding water, disrupting sediment transport, altering vertical hydraulic gradients and varying flow dynamics (Schmadel et al., 2016; Gerecht et al., 2011; Fritz and Arntzen, 2007; Arntzen et al., 2006). The daily stage fluctuation from hydroelectric dams for example, regulate the size of the hyporheic zone and the magnitude and frequency of HEF (Sawyer et al., 2012; Gooseff et al., 2006; Lautz et al., 2006; Harvey and Bencala, 1993b). In case of dam-induced water levels changes, a temporal lag occurs between stream stage and aquifer water;

HEF is transient and penetrates several meters into the riparian aquifer with residence times of hours (Sawyer et al., 2009). Schmadel et al. (2016) predicted HEF and residence times from the timing and magnitude of diel fluctuations and valley slope, and found that minimal exchange occurs when the magnitude of stream level fluctuations coincide with the hillslope water table, while maximum exchange occurs when stream stage is out of phase with the hillslope and therefore larger amplitude in stream and hillslope occur. Studies using thermal sensors have reported differences of HEF within the subsurface upstream and

downstream of dams, attributed to the overall hydraulic behavior around the dam and to the changes in topography induced by the dam (Hester et al., 2009; Fanelli and Lautz, 2008). Upstream and downstream pools create by ponding and channel degradation, respectively, have the potential to drive bedform-scale exchange flow. Temperature results suggest that highest hyporheic exchange rates occur downstream of dams, while HEF is limited in upstream pools where fine sediment deposits yield low hydraulic conductivities (Fanelli and Lautz, 2008).

## 7.3 Land management and use: impacts on sediment delivery, channel complexity and hydrological regime

Land cover and management impacts on HEF through several pathways, as it impacts on the quality (i.e. sediment delivery and channel complexity) and quantity (i.e. discharge, infiltration, evapotranspiration (ET)) of groundwater and surface water (Santos et al., 2015; Carrillo-Rivera et al., 2008). The relationship between land use, sediment delivery and HEF remains an area of active research, but in general both urbanization and agriculture significantly modify channel morphology, streambed sed-

iment size, and hydraulic conductivity by competing effects from increasing fine sediment inputs (which decrease streambed hydraulic conductivity) and stream discharge (which increases advective HEF) (Emanuel et al., 2014; Ryan et al., 2010; Kasahara and Wondzell, 2003; Morrice et al., 1997; D'angelo et al., 1993). First, decreased porosity and permeability of streambed sediments, e.g., due to increased sediment loads from agriculture, is usually connected to decrease of in channel storage and





hyporheic exchange flows (Packman and MacKay, 2003; Brunke and Gonser, 1997). Secondly, water abstraction often include both pumping stream surface and groundwater, which can increase groundwater levels and thereby increase groundwater discharge to streams and/or decrease stream water flow to groundwater (Winter, 1998). Lower water tables generally reduce the vertical extent of the HZ by increasing water losses from the stream and reducing the hydraulic gradients that drive HEF

(Hancock, 2002). Not only the magnitude but also the length of the hyporheic exchange flows are affected: tracer experiments conducted on several reaches within a single land use type showed a reduction of transient storage as a function of the surrounding land use due to lower geomorphological complexity in agricultural streams, promoting the formation of low-flow zones but reducing HEF (Gooseff et al., 2007). However, little research has been carried out on HEF in urban rivers where low morphological complexity and anthropogenic factors have impacted streams substrates and planforms (Drummond et al.,

2017; Gooseff et al., 2007; Grimm et al., 2005; Groffman et al., 2005; Walsh et al., 2005).

## 8   Case study: the River Tern

While previous sections described how individual factors influence HEF, these factors interact across spatial scales to produce a high degree of spatial and temporal heterogeneity in HEF. To illustrate the challenges in resolving hyporheic exchange across scales, we use the River Tern (UK) as a case study. We first review previously published research on HEF in this

stream, and then discuss the multi-scale factors that influence HEF based on the review presented previously in Sections 3 to 7. HEF has been studied in great detail at the sub-reach scale in the River Tern (Krause et al., 2013; Angermann et al., 2012; Krause et al., 2011b; Hannah et al., 2009). Results indicate that that spatial variations in surficial geology of the floodplain and temporal variations in groundwater levels control local river-aquifer interactions, and dictate the rates and patterns of HEF. Strong correlations between rainfall and groundwater levels indicate that the river acted as a recharge boundary, and pumping

tests suggest that hydraulic continuity of bedrock with the River Tern is greater at high flows than at low flows (Streetly and Shepley, 2005). At more local scales, Hannah et al. (2009) and Angermann et al. (2012) found that spatial heterogeneity in HEF is controlled by both topography and streambed strata. Heat tracer studies identified inhibition of hyporheic flow in peat and clay lenses below the stream (Angermann et al., 2012). Because of this structure, hyporheic flow paths in riffles did not coincide with the patterns expected from topography-induced head distributions, and instead seem to be driven by

locations of confining peat and clay strata Angermann et al. (2012). Temperature data indicated that advected surface water or groundwater control heat transport within the hyporheic zone (Hannah et al., 2009). Hannah et al. (2009) and Anibas et al. (2012) showed that the local hydrogeological and geomorphological context explains the observed seasonal thermal differences between riffles: increased downwelling at riffle tails during winter results from greater groundwater influence and high water stage (Fig. 4). These results highlight the need to integrate interpretations of observed rates and patterns of hyporheic exchange

with hydrogeological and geomorphological context. As a starting point, valley type can be used to predict the development and extent of lateral hyporheic exchange. We illustrate the generic nature of valley confinement for the River Tern considering the headwater valley of the Tern at Norton-in-Hales and including the 150 m reach considered in previous studies (Hannah et al., 2009). The catchment is low-lying, with average elevations between 50 and 120 m, and the area is predominantly





agricultural, with croplands and pastures accounting for the majority of the land area (Fuller et al., 2002). The valley section has an elevation ranging from 91 to 114 m, a low channel gradient between 0 and 0.2% and is laterally unconfined. The River Tern and its tributaries are underlain by Permo-Triassic sedimentary rocks (sandstone and conglomerate interbedded), which dominate river-aquifer interactions at regional scale (Allen et al., 1997). This permeable geology supports unconfined highly,

moderately-productive aquifers characterized by intergranular flows. However, most of the surficial geology of the catchment is from the Pleistocene age, ranging from sand and gravel to diamicton, peat and clay. The thickness varies spatially across the catchment, with thicker areas in the western part of the catchment comprising up to 30 m of till (Streetly and Shepley, 2005). Throughout the length of the selected section, the river is fringed by wet woodland, predominantly *Alnus glutinosa*. The bedrock is mainly sandstone and mudstone, whereas the superficial geology is sand and gravel with some silt, clay and

diamicton. The valley was divided into reach sections of 850 m and analyzed the confinement according to the framework of Fryirs et al. (2016). Some reaches are laterally constrained by anthropogenic structures (roads, houses) in one or both sides (Table 2, Fig. 5). The anthropogenic confinement is most prominent in proximity to the town, where the active floodplain is artificially disconnected by engineered structures. Given that the channel planform is mostly meandering, and is not constrained by bedrock (Section 4.5), lateral hyporheic flows will likely occur predominately in unconfined areas, where the planform can

adjust to its sinuous-meandering shape (i.e. reaches 1, 4, 5, 6, 7, 8, 9, 10 in Table 2). According to the hydrogeology of the area (Section 5.1), hydraulic conductivities are expected to be highly variable as consequence of the sediment sorting and HEF will likely vary within reaches when arenaceous and rudaceous lithologies dominate on argillic and peat sediments (i.e. reaches 2, 4, 5, 6, 8, 9 in Table 2). Finally, differences along the general gradient of the network (Sections 4.6 and 4.7) are expected where the conjunction of increase of riverbed slope, meander bends, and bedforms (Section 4.3) will likely increase hydraulic

head gradients and induce HEF (i.e. reaches 4, 5, 7, 8 in Table 2). Previous research suggests that the mosaics of hyporheic exchange in the River Tern are induced by spatial variations in streambed topography and sediment permeability and temporal variations in groundwater recharge. Through the discussion of this case study, we illustrated that assessment of the geological and morphological context for the river channel can help to explained observed patterns in bedform-driven HEF. This work outlines the opportunity to build HEF scaling relationships from basic patterns of channel morphology, valley confinement,

and hydrogeological properties.

## 9  Conclusions

Information on the underlying drivers of HEF across space and time, and unravelling the process interactions between them, is essential to predicting HEF patterns in catchments. However, we are currently unable to fully capture the extent of the interaction between factors that drive HEF. This review highlighted the factors operating over multiple spatial and temporal scales that govern HEF, and summarise how they interact to determine HEF. Predictive relationships are needed to enable upscaling

to catchment scales or downscaling to sub-reach-scales, as well as the response of HEF to changing hydrological, topographical, geological, ecological and anthropogenic conditions. The ability to understand the temporal and spatial dynamics of HEF depends on the holistic perspective suggested here, which considers co-variations between flow, slope, valley confinement,





catchment area, sediment size, and river planform and bedforms morphology. Direct data on HEF at larger scale than reaches are severely limited. By summarizing the factors responsible for rates and patterns of HEF in river systems this review provides a comprehensive understanding and evaluate the characteristics of hyporheic flows in conjunction with and embedded within catchment and valley characteristics.

5    *Author contributions.*  C. Magliozzi developed and prepared the manuscript with contribution from all co-authors. All authors have approved the final article

*Competing interests.*  The authors declare that they have no conflict of interest

*Acknowledgements.*  This work was supported by the Marie Skłodowska-Curie Action, Horizon2020 within the project HypoTRAIN. Grant agreement no: 641939. A. I. Packman was also supported by U.S. NSF grant EAR-1344280.





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



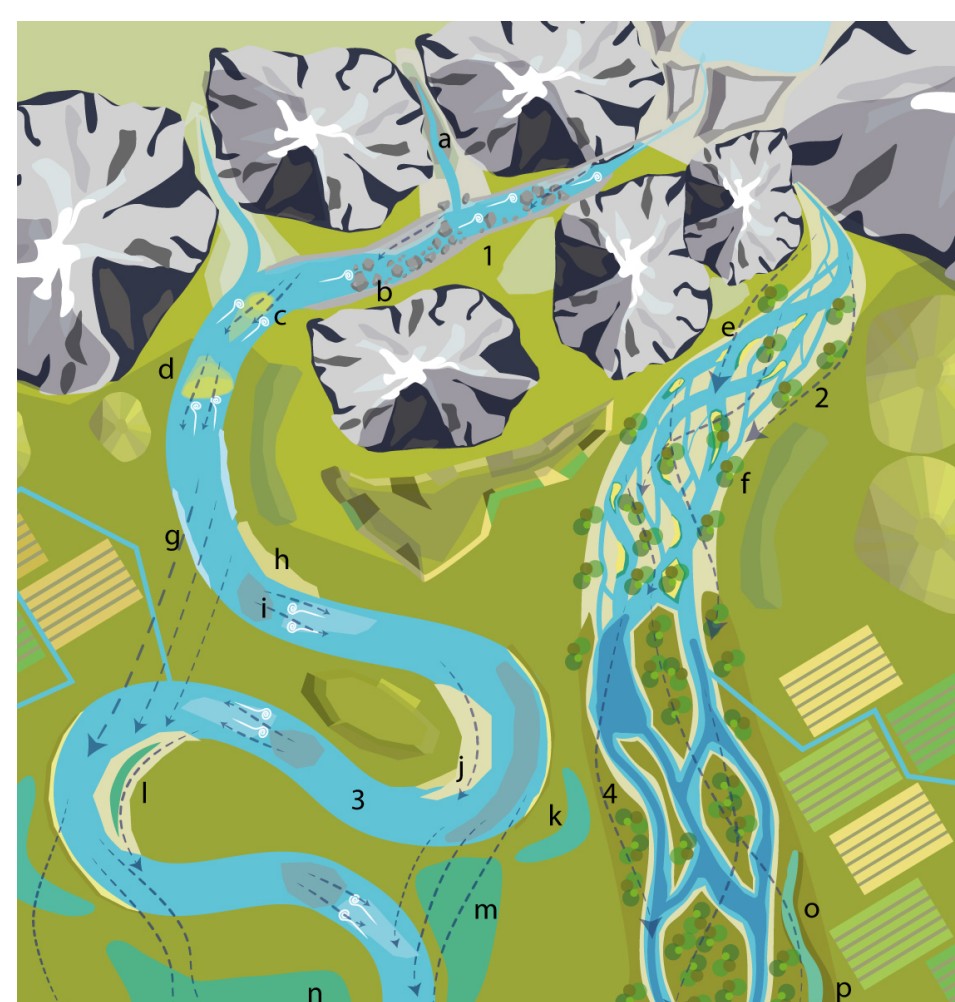

**Figure 1.** Illustration of catchment complexity: scales and features that influence hyporheic exchange flows. Spatial changes in surface topography, land use and vegetation, drive geomorphological and hydrological changes at valley and reach scale. At catchment scale, variations in surface topography shapes valleys and channel types. Feature 1 refers to confined valleys characterized by straight channels, meandered and braided, and the following floodplain features: scour holes and gravel splays (a). The straight channel presents in-channel cascades (b) geomorphic features. Feature 2 refers to braided channel morphology with multi-thread channel, an undulating floodplain of bars and islands. In-channel geomorphic units are several types of bars (e), such as mid and lateral bars, and vegetated islands (f). Feature 3 represents a sinuous-meandering floodplain with occasional oxbow lakes and backwater swamps (m, n, k) and in channel: longitudinal bar (c), transverse bar (d), counterpoint bar (h), pond-riffle (i), point bar (l), chute channel (j). Feature 4 indicates an anabranching valley with multi-thread channels including abandoned channels (o) and backwater swamps (p). The channel can be quite deep and include islands covered with vegetation. The symbol * refers to irrigation system of the adjacent agricultural fields.





**Figure 2.** Conceptual diagram of the key drivers of the hyporheic exchange across scales. This diagram can be read from the centre to the outer part and viceversa as indicated by the black arrows. Dashed lines represent hidden boundauries between scales. Color gradient, from light to dark, follows the hierarchical approach of this review from channel-scale to reach-scale to catchment-scale.





**Figure 3.** Representation of channel planforms. Sinuosity influences water exchange within a river segment. Hyporheic exchange increases with sinuosity due to hydraulic gradients in the meander neck (Section 4.6).



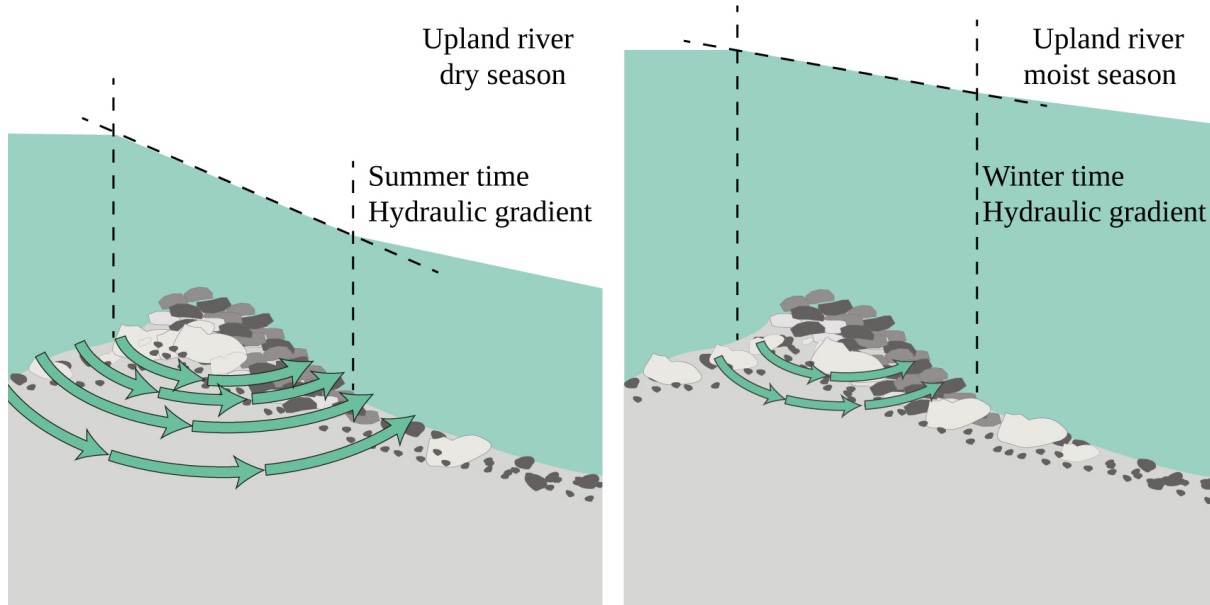

**Figure 4.** Conceptual representation of seasonal variation of hydraulic gradient with water stages in a upland environment. Development of hyporheic exchange in a riffle considering extension and contraction of hyporheic sediment.





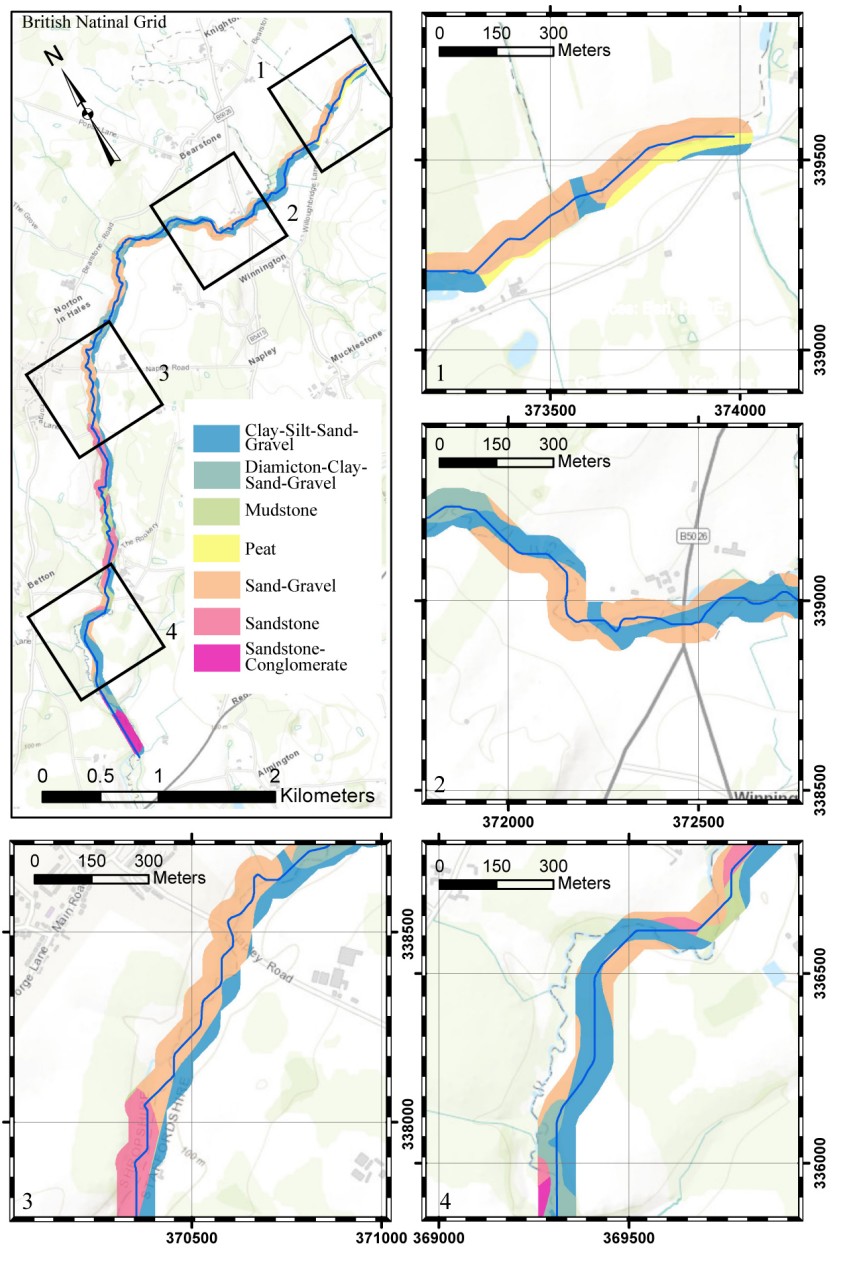

**Figure 5.** Examples of reaches of the River Tern analysed for HEF (Table 2). The river is subdivided into reaches based on their planform morphology (sinuosity units: when the overall direction of the planimetric course changes) and classified in Table 2. The figure represents for each reach, the main river, surficial and bedrock geology in a buffer area of 50 m from the main channel. Surficial and bedrock geology are represented as greater the connectivity within sediment strata and higher the HEF. Vertical HEF will be restricted by low permeability units and unconsolidated deposits and lateral HEF by grain size material, river sinuosity and cross-valley head gradients.





**Table 1.** Expected hyporheic exchange flows in different channel types. Geomorphic unit refers exclusively to in-channel features. Sources: Gurnell et al. (2016); Wondzell and Gooseff (2013); Buffington and Tonina (2009); Nanson and Croke (1992)

| Floodplain type | Characteristics floodplain | Floodplain geomorphic units | Expected HEF floodplain scale | Channel planform type | Channel geomorphic units | Expected HEF channel |
|---|---|---|---|---|---|---|
| Confined, steep, narrow valley | High energy, coarse sediment from poorly sorted, boulders and gravel with sand and soils to basal gravels with sand and silt. | Large boulder, levee, deep black channels, scour holes | Confined HEF which is driven by in-channel geomorphic units. Short residence times. | Colluvial: single-threat or straight sinuous. | Rock-steps, cascades, rapids. | Colluvial, channels are usually strongly confined and stable. Very coarse and supply-sediment limited. Cascade and rock steps would likely allow head variations and near-bed turbulence which enhance HEF. |
| | | | | Alluvial: single - threat or straight sinuous. | Small pools, step-pools, bars. | Alluvial channels are relatively stable for long (very coarse to coarse material).Broken,fast-flowing turbulent flow. Where step-pool units occur the HEF is enhanced. |
| Braided confined, partly confined, unconfined | Medium energy. Abundant sediment load from gravel to sand and silt | Abandoned channels, bars, islands. | Lateral HEF enhanced by alluvium and head variations (hydraulic conductivity). High residence time | Alluvial channels. Multi-thread | Pools, riffles, riffle-pools, laterals bars, mid-channel bars, islands. | Highly instable vertically and laterally. Sediment supply high. Expected head variations, alluvial volume variations and hydraulic conductivity variations that likely enhance HEF. |
| Sinuous, meandering | Medium energy. Mostly sand with silt and gravel | Smooth to undulating, floodplain surface often with areas of vertically accreted fine sediments , backswamps, ponds | Lateral HEF enhanced by alluvium and head variations (hydraulic conductivity) or limited by fine deposited areas. High residence time. | Single, thread-sinuous or meandering | Pools, riffles, point bars, bars, dunes. | Relatively unstable and subjected to progressive migration. The instability reflects the geomorphologic units that likely promote HEF. |
| Anabranching | Low energy. Fine sediments from silts and clays to sands | Flat, floodplains, extensive islands, peat and lakes swamps, splayes, side-levee | Vertical HEF enhanced or limited by alluvium and head variations. | Multi-thread anabranching | Islands, ripples and dunes, abandoned channels. | Predominantly, stable. Channel variation in sediment alluvium would likely drive some vertical HEF. |



**Table 2.** Case study about the river Tern, UK (Section 8). The table describes the 10 reaches sections obtained by dividing the river channel into sinuosity units based on changes in the axis of the overall planimetric course. The units that differed in sinuosity by more than 10% were considered separate reaches. Surface geology and valley type are evaluated with respect to the extent of lateral hyporheic exchange. The sections are enumerated and described from upstream to downstream. Information of geology extracted from the British Geological Survey website.

| Reaches | Underlying geology | In-channel sediment | Description | Channel Gradient(%) | Sinuosity |
|---|---|---|---|---|---|
| 1 | Sandstone-conglomerate bedrock of Triassic period. Surficial geology, sedimentary substrate of quaternary period. Alluvial, fluvial and glacigenic sediments | Min grain is clay. Max grain is gravel. Mixed argillic and arenaceous grains. | Unconfined valley on both banks. The river is meandering and the riparian vegetation is abundant | 0.001 | 1.089 |
| 2 | Mudstone and sandstone bedrock of Triassic period. Surficial geology, sedimentary substrate of quaternary period. Alluvial and glaciofluvial sediments. | Predominant min grain is sand and max grain is gravel. Dominant grain is sand. Arenaceous – rudaceous grains. | Partially confined valley due to industrial plants and homes on the right bank of the river. The river is sinuous with the presence of a big meander and abundant riparian vegetation | 0 | 0.487 |
| 3 | Bedrock: mudstone and sandstone interspersed. Sedimentary geology of Triassic period. Dominance of fluvial sediments. | Min grain mud and clay and max grain is gravel. Dominant grains sand and mud. Argillic –rudaceous grains. | Partially confined valley due to homes on the right bank of the river. The river is overall sinuous with the presence of small meander and very abundant riparian vegetation | 0.052 | 0.537 |
| 4 | Bedrock: sandstone. Surficial geology, sedimentary substrate of triassic period. Dominance of fluvial deposits. | Min grain is mud, max grain is gravel. Dominant grain is sand. Arenaceous –rudaceous grains. | Mostly unconfined valley, presence of homes on the right bank of the river.The river is meandering and abundant riparian vegetation | 0.261 | 1.962 |
| 5 | Surficial geology of quaternary period. Dominance of glaciofluvial deposit. | Min grain is clay, max grain is gravel. Dominant grain is sand. Arenaceous –rudaceous grains. | Mostly unconfined valley, presence of homes on the left bank of the river. The river is forming small meanders and abundant riparian vegetation | 0.03 | 0.718 |
| 6 | Surficial geology of quaternary period. Dominance of glaciofluvial and glacigenic deposit. | Min grain is clay, max grain is gravel. Dominant grain is sand. Arenaceous –rudaceous grains. | Unconfined valley on both banks. The river is meandering and riparian vegetation is present throughout its length but mostly on the left bank. | 0.011 | 0.6 |
| 7 | Surficial geology of quaternary period. Dominance of glaciofluvial and glacigenic deposit. | Min grain is clay, max grain is gravel.Dominance of clay with gravel. Mixed argillic andrudaceous grains. | Unconfined valley on both banks presence of a small bridge. The river is meandering and riparian vegetation is present throughout its length although more scarce with comparison to previous sections. | 0.06 | 1.87 |
| 8 | Surficial geology of quaternary period. Dominance of glaciofluvial and alluvial deposit. | Min is clay and max is gravel. Mixed arenaceous and argillic grain. | Mostly unconfined valley, presence of industrial plant on the left bank of the river. On the left bank there are two ponds. The river is forming small meanders, riparian vegetation is present. | 0.05 | 1.06 |
| 9 | Surficial geology of quaternary period. Dominance of glaciofluvial and alluvial deposit. | Min is clay and max is gravel Predominance of sand grains. | Unconfined valley on both banks. The river is meandering and riparian vegetation is present and abundant on the left bank. Presence of pond. | 0.003 | 0.943 |
| 10 | Surficial geology of quaternary period. Dominance of glaciofluvial and fluvial deposit. | Min grain is clay ad max grain is gravel with presence of silt as well. Mix of arenaceous and rudaceous grains with peat and argillic. | Unconfined valley on both banks. The river is mostly sinuous and riparian vegetation is abundant on both banks. | 0.012 | 0.826 |