# Peer review of "Toward a conceptual framework of hyporheic exchange across spatial scales"

_Hydrology and Earth System Sciences, 2018_

## Short Comment (SC1) · 1 Jun 2018

This paper faces a very complex issue from a modelling point of view. There is no "a priori" formalisation of the problem and no clue on the features to use to numerically describe it. Even the possibility to "quantify" the existence of the hyporheic exchange is questionable. Thus, this is one of the most difficult situations for a classification problem. In this view, the analysis and the methodology reported in the paper is inventive, valuable, and convincing IMHO.

---

## Referee Comment (RC1) · Anonymous Referee #1 · 13 Jul 2018

General comments

The paper addresses relevant scientific questions on the understanding of process-based interactions between factors operating at different spatial and temporal scales driving hyporheic exchange flows (HEF). Understanding of HEF is important for many applications, for instance, for predicting the fate of contaminates in the surface environment in safety assessments of geological waste disposals. The paper is well written and suits the Journal well. I like the way that the review paper is presented based on various drivers. This gives reader a clear picture of which factors affect HEF.

Specific comments

It might be worth to point out why rivers are not isolated systems but interact continuously with groundwater in the Introduction. This may be obvious to the authors but may not be so for all readers. Groundwater discharge points generally coincides with topographic lows in the landscape, such as streams, lakes and wetlands (e.g. Marklund et al., 2008). With the seasonal variation, such discharge points can also change with time to become recharge points, and similar temporal fluctuations in the hyporheic zone and large-scale groundwater circulation is not discussed in this paper. An important driver for hyporheic flows are the static and dynamic pressures as discussed by the authors. However, what is actually the difference between dynamic and hydrostatic head gradients around channel morphological elements? A dynamic head (say velocity head) is gradually transformed to a static head along a stream-line that approaches a stagnation point at the bed. The pressure at the stagnation point is also affected by the static head defined by the water surface topography. In the end, the subsurface flow is driven (mostly, in a linear or Darcy flow theory) by static head gradients and this distinction is not so clear in the paper. The further research areas/topics might be worth to highlight in the Conclusions again.

Minor Corrections

Page 2, row 4: floadplain should be floodplain

Page 2, row 6: vertically and laterally (i.e. flood spates, overbank flows, etc.; (Minshall et al., 1985; Newbold et al., 1982, 1981), should be vertically and laterally i.e., flood spates, overbank flows, etc. (Minshall et al., 1985; Newbold et al., 1982, 1981);

Page 13, row 31: water consumption (i.e. ET) should be evapotranspiration (i.e. ET)

Page 16: row 25: . . . strata Angermann et al., (2012). should be . . . strata (Angermann et al., 2012).

Reference

Marklund, L., Wörman, A., Geier, J., Simic, E., Dverstorp, B., 2008. Impact of landscape topography and quaternary overburden on the performance of a geological

repository of nuclear waste. Nuclear Technology, 163, 165-179.

Please also note the supplement to this comment:
https://www.hydrol-earth-syst-sci-discuss.net/hess-2018-268/hess-2018-268-RC1-supplement.pdf

---

## Author Comment (AC1) · 17 Jul 2018

Dear Referee, Thank you for spending the time to review the manuscript and provide detailed feedback. We are glad that you liked the structured presentation of HEF drivers and can see their importance and relevance to environmental research and applications. We agree with the suggested revisions that you outlined in the specific comments section, and feel that added explanation and clarification in selected sections would help to improve the quality of our manuscript. Please find below a preliminary response to your main points. We agree with you that the concept of rivers continuously interacting with groundwater should be clarified to emphasize the impact of topography, at all scales, on spatial and temporal variations of groundwater flows. Therefore, we will discuss this aspect in Section 1 following lines 5-8 in page 2, by

introducing that i) the differences in hydraulic potential created by topography drive groundwater flows (i.e. discharge patterns follow topographic lows) and ii) the relative importance of continental groundwater flows and local discharge areas on rivers would depend on both large- and small-scale topography as demonstrated by spectral analysis. Concerning the definition of hydrostatic and hydrodynamic head gradients around channel morphological elements, we agree that an additional explanation is needed to clarify the differences in this manuscript. We believe the best section to address this point in Section 4.1 where HEF is discussed at the scale of in-channel bedforms, and through cross-referencing the previous sections.

Finally, the conclusion section will include references to field applications (i.e. river restoration management) in order to highlight the need for a process-based understanding of river-groundwater interactions to river management.

Page 2, row 4: floadplain should be floodplain

Yes, we will correct it

Page 2, row 6: vertically and laterally (i.e. flood spates, overbank flows, etc.; (Minshall et al., 1985; Newbold et al., 1982, 1981), should be vertically and laterally i.e., flood spates, overbank flows, etc. (Minshall et al., 1985; Newbold et al., 1982, 1981);

Yes, we will correct it

Page 13, row 31: water consumption (i.e. ET) should be evapotranspiration (i.e. ET)

Yes, we will correct it

Page 16: row 25: . . . strata Angermann et al., (2012). should be . . . strata (Angermann et al., 2012).

Yes, we will correct it

A double check on the full document will also be performed.

---

## Short Comment (SC2) · 22 Aug 2018

This study addresses an essential aspect common to understanding many natural systems, namely the spatiotemporal variation and interaction of multiple controlling factors. Specifically, the paper presents a systematic and extensive review on hydrological, topographical, hydrogeological, ecological and anthropogenic factors controlling spatial and temporal variation in hyporheic exchange flows in rivers. The authors clearly demonstrate the importance of considering interactions between multiple drivers to understand and predict variations in the hyporheic exchange flows. I therefore think the study offers a useful and holistic framework for future studies on the hyporheic zone, which will especially guide data collection and predictive modelling.

---

## Short Comment (SC3) · 22 Aug 2018

Dear K. Vercruysse, thank you for commenting on the manuscript. I am glad you can see its relevance to hyporheic research and applications.

---

## Short Comment (SC4) · 28 Aug 2018

In my opinion, this manuscript is a good step stone for those who want to further explore the processes taking place in the hyporheic zone and their relationship with different spatial scales within a catchment. The hyporheic zone is probably one of the least understood hydrological concepts, and it is often disregarded by water managers and practitioners outside academia. However, the hyporheic zone and the processes occuring within it are key to many other hot topics in the field such as water quality, water bodies ecological status, or evaluation of ecosystem services.

Achieving a better understanding of the hyporheic zone can lead to better and focused measures definition, improving overall environmental status of river basins, and reduc-

ing implementation costs.

---

## Short Comment (SC5) · 8 Sep 2018

Dear D. Haro-Monteagudo, thank you for your comment on the manuscript. I am glad you have pointed out the relevance of this research to a broader management context. Holistic approaches, at the interface of river hydrology, geomorphology, and ecology, have been advocated to keep river management open and flexible to the challenges of maintaining the natural functioning of rivers in current times of changes. River management is, in fact, adopting more often catchment-based approaches. The factors presented in this manuscript are not only important for HEF but to river processes in general. Structuring the review around multiple scales improves spatial and temporal understanding of the variability of environmental factors in river systems and how reaches have been impacted by catchment-scale changes. Therefore, this manuscript

underscores broader river management planning that includes catchment-scale solutions. We will give perspectives on future research areas in this direction in the Conclusion section.

---

## Referee Comment (RC2) · Anonymous Referee #2 · 25 Sep 2018

General comments This is an extremely detailed review paper that addresses the scientific community's relevant and current scientific understanding of the drivers of hyporheic exchange flows across spatial and temporal scales. The authors do a thorough job demonstrating the rationale and importance of different controls on hyporheic exchange flows. Overall, this review-style manuscript is suitable for HESS and would contribute nicely to the scientific body of literature after a thorough read-through and edit by the authors to address word choice, grammar and typo issues. Currently, although the manuscript does a great job summarizing the state of the science, it suffers from a range of typo, missing words, and incorrect forms of grammar that distract from the manuscript's message. Further, some of the explanations are not clear and would benefit from more descriptive terminology. See the Specific Comments section for sev-

eral examples.

Specific comments 1. There are a range of typos, incorrect forms of grammar. I marked some of them in the technical corrections section, but please take care to read through the manuscript carefully.

2. Some of the explanations are not clear. More descriptive terminology would help with this. I provided some examples of where terminology is not clear, but I encourage the authors to read closely through the manuscript to address all confusing terminology. Title for Section 3.2 is confusing – in a large scale what? Also, what does "large" refer to? Perhaps catchment scale would be more descriptive? Though, after reading this section, it seems this section is encouraging researchers to take the context of the landscape into consideration for reach-scale research. Thus a more suitable title may be something like: "Reach scale HEF in the context of the larger landscape" P 8 L 1: The language in this sentence is a little unclear. Please clarify what "water table is continuous on the bedrock" means. P L 22: This sentence is confusing - As valley confinement what? Perhaps the authors meant "As similar to valley confinement,". Please clarify language.

3. Section 4.2 L 22-24: Quite a bit of field research has been conducted around partially submerged bedforms from the Lautz research group, specific to investigating HEF around restoration structures that mimic natural bedforms. See Gordon et al.2013, Lautz and Fanelli, 2008, Zimmer and Lautz, 2014 for examples.

4. Is there a citation that can back up the P 12 L 6-7 statement?

5. Section 7.2: It should be noted that dams (and many of the anthropogenic changes) can occur on small (Fanelli and Lautz, 2008) and much larger sized (Fritz and Arntzen, 2007) rivers. It would be interesting to explore the relative control of dams on such different sized systems, relative to channel slope, etc.

Technical corrections P 2 L 2: remove first "and" P 2 L 3: Replace "HEF but less" to

"HEF, but are less" P 2 L 18: Add "the" before "catchment scale" P 3 L 25 – Change reach scale to "the reach scale" or "reach scales". There are several instances like this throughout the manuscript, please address them all. E.g. P 3 L 28: change "HZ" to "the HZ". P4 L 21 – Missing a word after "at a larger" P 5 L 21 – change "though" to "through" P 7 L 21: change "his study" to "this study" or "their study".

---

## Author Comment (AC2) · 30 Sep 2018

Dear Referee,

Thank you for reviewing the manuscript and providing your detailed feedback. We are glad that you recognize its relevance to hyporheic research and applications, and its suitability for HESS journal. We agree with the suggested revisions and recognise that clarifications in the selected sections would help to improve the message and the quality of our manuscript. Please find below point-to-point responses to your main points.

Specific comments 1. There are a range of typos, incorrect forms of grammar. I marked some of them in the technical corrections section, but please take care to read through

[Figure]

the manuscript carefully. 2. Some of the explanations are not clear. More descriptive terminology would help with this. I provided some examples of where terminology is not clear, but I encourage the authors to read closely through the manuscript to address all confusing terminology.

Yes, we will read carefully through the manuscript and correct typos, grammatical forms and address where terminology is not clear.

Title for Section 3.2 is confusing – in a large scale what? Also, what does "large" refer to? Perhaps catchment scale would be more descriptive? Though, after reading this section, it seems this section is encouraging researchers to take the context of the landscape into consideration for reach-scale research. Thus a more suitable title may be something like: "Reach scale HEF in the context of the larger landscape"

Yes, we agree. We will change the title with "Reach scale HEF in the context of the larger landscape"

P 8 L1: The language in this sentence is a little unclear. Please clarify what "water table is continuous on the bedrock" means.

The sentence "For example, HEF transport is expected to be more uniform when the water table is continuous on the bedrock than when the water table falls and interacts directly with bedrock topography (Ward et al., 2012)" will be modified in : "For example, HEF transport is expected to be more uniform in lowland rivers, where the flat land surface and shallow aquifers with low transmissivity favour a topographically-controlled water table, than in upland environments where bedrock outcrops may confine HEF and influence cross-valley hydraulic gradients (Ward et al., 2012)"

P8 L 22: This sentence is confusing - As valley confinement what? Perhaps the authors meant "As similar to valley confinement,". Please clarify language.

Yes, we will modify with "As with valley confinement".

3. Section 4.2 L 22-24: Quite a bit of field research has been conducted around partially submerged bedforms from the Lautz research group, specific to investigating HEF around restoration structures that mimic natural bedforms. See Gordon et al. 2013, Lautz and Fanelli, 2008, Zimmer and Lautz, 2014 for examples.

Yes, we acknowledge that and we will modify Section 4.2 Lines 22-24. "Current knowledge of hyporheic fluxes and their spatio-temporal variability in submerged bedforms has been obtained from simulations (Boano et al., 2014; Irvine et al., 2014; Trauth et al., 2014; Stonedahl et al., 2013; Janssen et al., 2012; Cardenas and Wilson, 2007; Elliott and Brooks, 1997), laboratory (Fox et al., 2014; Tonina and Buffington, 2007) and field experiments (Zimmer and Lautz, 2014; Gordon et al. 2013; Lautz and Fanelli, 2008)."

4. Is there a citation that can back up the P 12 L 6-7 statement?

Yes, there are citations to support the statement. For example Kunz et al. (2017), Sun et al. (2015), and Gooseff et al. (2007).

The following references will be added to the reference list:

Gooseff, M. N., Hall, R. O., and Tank, J. L.: Relating transient storage to channel complexity in streams of varying land use in Jackson Hole, Wyoming, Water Resources Research, 43, 2007.

Sun, N., Yearsley, J., Voisin, N., & Lettenmaier, D. P.: A spatially distributed model for the assessment of land use impacts on stream temperature in small urban watersheds, Hydrological Processes, 29(10), 2331-2345, 2015.

Kunz, J. V., Annable, M. D., Rao, S., Rode, M., & Borchardt, D.: Hyporheic passive flux meters reveal inverse vertical zonation and high seasonality of nitrogen processing in an anthropogenically modified stream (Holtemme, Germany), Water Resources Research, 53(12), 10155-10172, 2017.

5. Section 7.2: It should be noted that dams (and many of the anthropogenic changes) can occur on small (Fanelli and Lautz, 2008) and much larger sized (Fritz and Arntzen,

2007) rivers. It would be interesting to explore the relative control of dams on such different sized systems, relative to channel slope, etc.

Yes, we agree. This is an interesting point to develop and we will add the following paragraph:

P15, L19: after "...occur." "As a consequence, the effects of dams on HEF vary with channel planform and streambed topography. For example, in river systems characterized by large alluvial channels and unconfined aquifers, the relationship between dam-induced changes in river stage and HEF is characterized by hysteresis (Fritz and Arntzen, 2007). Therefore, HEF is not only dependent on changes in river stage but also on the difference between river and aquifer elevations (Fritz and Arntzen, 2007). As river stage varies, there is a fast response of hyporheic flows which rapidly change with the head difference within the HZ, and a slower response of HEF with changes in elevation head of the near aquifer (Fritz and Arntzen, 2007). Additionally, the lower hydraulic conductivity near the surface of the HZ, caused by accumulation of sediment in the alluvial matrix and often characterizing alluvial channels (Section 5.2), might restrict the changes in hydraulic pressure over the first cm of river sediment (Fritz and Arntzen, 2007). In river systems characterized by small channel sizes and complex streambed morphology, differences of HEF within the subsurface upstream and downstream of dams have been attributed to the overall hydraulic behaviour around the dam and to the changes in topography induced by the dam (Hester et al., 2009; Fanelli and Lautz, 2008). Studies using thermal sensors have reported that upstream and downstream pools created by ponding and channel degradation, respectively, have the potential to drive bedform-scale exchange flow. Temperature results suggest that the highest hyporheic exchange rates occur downstream of dams, while HEF is limited in upstream pools where fine sediment deposits yield low hydraulic conductivities (Fanelli and Lautz, 2008)."

Technical corrections

P 2 L 2: remove first "and" Yes, "and" will be removed

P 2 L 3: Replace "HEF but less" to "HEF, but are less" Yes, will add "," after "HEF"

P 2 L 18: Add "the" before "catchment scale" Yes, will add "the"

P 3 L 25 – Change reach scale to "the reach scale" or "reach scales". There are several instances like this throughout the manuscript, please address them all. Yes, we will change "the reach scale" accordingly throughout the manuscript.

E.g. P 3 L 28: change "HZ" to "the HZ". Yes, we will change "HZ" to "the HZ".

P4 L 21 – Missing a word after "at a larger" Yes, we will add "at a larger scale (i.e. catchment)".

P 5 L 21 – change "though" to "through" Yes, we will change "though" with "through"

P 7 L 21: change "his study" to "this study" or "their study". Yes, we will change "his study" with "their study"

In addition to the typos and grammatical errors already identified, we will:

P2, L2: change "turbulence" in "turbulence". Remove "and". P2, L4: change "floadplain" in "floodplain". P2, L7: correct "vertically and laterally (i.e. flood spates, overbank flows, etc.; (Minshall et al., 1985; Newbold et al., 1982, 1981)" with "vertically and laterally i.e., flood spates, overbank flows, etc. (Minshall et al., 1985; Newbold et al., 1982, 1981)" P3, L25: change "at reach scale" with "at the reach scale" P4, L5: change "the HZ was found" with "the HZ has been found" P4, L10: invert "affect significantly" with "significantly affect" P5, L4: invert "have usually" with "usually have" P5, L31: change "into" with "in" P6, L14: remove "s" in "times" P6, L18: add "the" before "longitudinal" P6, L28: add "the" before "riffle-pool" P9, L 22: remove "forefront" with "priority" P9, L 23: add "the" before "stream" P9, L 27: add "a" before "result" P12, L21: remove "Certainly" and capitalize "In-channel". Remove "the" in front of "create" P15, L21: change "create" with "created" P16, L17: removed "that" that is typed twice

P16, L23: replace "Because of" with "Given" Subheading 4.2: correct "an in-channel bedforms", with "in-channel bedforms"

---

## Author Response (AR1)

Dear Editor,

Please find below our point-by-point response (in red) to the reviewers comments.
A marked-up manuscript version of the .pdf is provided with changes highlighted in yellow. As suggested, besides the adjustments required by the referee, we have read carefully through the manuscript and corrected typos, grammatical forms and addressed where terminology was not clear. This was also marked-up and reported at the end of this file.

**Point-by-point response to reviewer #1 (13/07/2018)**

P 2, L 4: "floadplain" changed in "floodplain".

P 2, L 6: "vertically and laterally (i.e. flood spates, overbank flows, etc.; (Minshall et al., 1985; Newbold et al., 1982, 1981)" changed to "vertically and laterally i.e., flood spates, overbank flows, etc. (Minshall et al., 1985; Newbold et al., 1982, 1981)," . (P2 L8)

P 13, L 31: "water consumption (i.e. ET)" changed to "evapotranspiration (i.e. ET)". (P14, L24)

P 16: L25: "… strata Angermann et al., (2012)" changed to "… strata (Angermann et al., 2012)" (P17, L28).

It might be worth to point out why rivers are not isolated systems but interact continuously with groundwater in the Introduction. This may be obvious to the authors but may not be so for all readers. Groundwater discharge points generally coincides with topographic lows in the landscape, such as streams, lakes and wetlands (e.g. Marklund et al., 2008). With the seasonal variation, such discharge points can also change with time to become recharge points, and similar temporal fluctuations in the hyporheic zone and large-scale groundwater circulation is not discussed in this paper.

We agree with the referee that the concept of rivers continuously interacting with groundwater should be clarified to emphasize the impact of topography, at all scales, on spatial and temporal variations of groundwater flows.

Section 1, P 2  L 5-10, changed :
"Catchment and river characteristics vary markedly along river networks affecting the groundwater and surface water flows that drive HEF. These variations include: (i) the differences in hydraulic potential created by topography-drive groundwater flows (i.e. discharge patterns follow topographic lows) and the temporal and spatial scales of the stream system from upstream to downstream, vertically and laterally i.e., flood spates, overbank flows, etc. (Minshall et al., 1985; Newbold et al., 1982, 1981); (ii) continental groundwater flows and local discharge areas on rivers  depend on both large- and small- scale topography as demonstrated by spectral analysis (Marklund et al., 2008; Wörman et al., 2007; Wörman et al., 2006); and (iii) complex geomorphological structures (armoring, bedforms, bars and other lateral variability within channels, braiding, meanders, floodplain deposits etc…)."

Added reference to list:
Marklund, L., Wörman, A., Geier, J., Simic, E., Dverstorp, B.: Impact of landscape topography and quaternary overburden on the performance of a geological repository of nuclear waste, Nuclear Technology, 163, 165-179, 2008.

An important driver for hyporheic flows are the static and dynamic pressures as discussed by the authors. However, what is actually the difference between dynamic and hydrostatic head gradients around channel morphological elements?

A dynamic head (say velocity head) is gradually transformed to a static head along a stream-line that approaches a stagnation point at the bed. The pressure at the stagnation point is also affected by the static head defined by the water surface topography. In the end, the subsurface flow is driven (mostly, in a linear or Darcy flow theory) by static head gradients and this distinction is not so clear in the paper.

P5, L 26-29:
"....HEF is proportional to the hydraulic head gradients in the streambed. Both hydrodynamic and hydrostatic forces generated by in channel bedforms have large effects on the variability of HEF from cm to m scale. However, in reaches where stream velocities are low relative to topographic variability, HEF will be mostly driven by hydrostatic head gradients defined by the water surface topography."

The further research areas/topics might be worth to highlight in the Conclusions again

In the conclusion Section 9, we added and changed the text as follows:

"Information on the underlying drivers of HEF across space and time, and their processes interactions is essential to predicting HEF in river networks. This review assembled, for the first, studies on drivers of HEF across multiple spatial and temporal scales, to provide a comprehensive overview of the mechanisms by which HEF is generated and modified via interactions between processes.
HEF plays such a significant role in mediating physical, chemical and ecological processes in rivers that considering the HZ in management plans could bring major benefits to re-establish the processes necessary to support the natural ecosystem within a catchment. But, the ability to understand the temporal and spatial dynamics of HEF depends on the holistic perspective suggested here, which considers co-variations between flow, slope, valley confinement, catchment area, sediment size, and river planform and bedforms morphology. Direct data on HEF at larger scale than reaches are severely limited and is required to improve methodological and modelling approaches to HEF and target river management needs (Magliozzi et al., 2018).
By summarizing the factors responsible for rates and patterns of HEF in river systems this review provides a comprehensive framework which support process-based hydroecological knowledge of HEF and the development of trasferable approaches to guide river management including the HZ in their prioritization and planning (Magliozzi et al., 2018)."

Added reference to list:
Magliozzi, C., Coro, G., Grabowski, R., Packman, A. I., & Krause, S.: A multiscale statistical method to identify potential areas of hyporheic exchange for river restoration planning, Environmental Modelling & Software, 2018.

**Point-by-point response to reviewer #2 (25/09/2018)**

Title for Section 3.2 is confusing – in a large scale what? Also, what does "large" refer to? Perhaps catchment scale would be more descriptive? Though, after reading this section, it seems this section is encouraging researchers to take the context of the landscape into consideration for reach-scale research. Thus a more suitable title may be something like: "Reach scale HEF in the context of the larger landscape"

Title changed to "Reach scale HEF in the context of the larger landscape"

P 8 L1: The language in this sentence is a little unclear. Please clarify what "water table is continuous on the bedrock" means.

Changed to (P 8 L16-19):
"For example, HEF transport is expected to be more uniform in lowland rivers, where the flat land surface and shallow aquifers with low transmissivity favour a topographically-controlled water table, than in upland environments where bedrock outcrops may confine HEF and influence cross-valley hydraulic gradients (Ward et al., 2012)"

P8 L 22: This sentence is confusing - As valley confinement what? Perhaps the authors meant "As similar to valley confinement,". Please clarify language.

Changed to (P 9 L7): "As with valley confinement".

3. Section 4.2 L 22-24: Quite a bit of field research has been conducted around partially submerged bedforms from the Lautz research group, specific to investigating HEF around restoration structures that mimic natural bedforms. See Gordon et al. 2013, Lautz and Fanelli, 2008, Zimmer and Lautz, 2014 for examples.

Yes, we acknowledge that and we modified Section 4.1.1 Lines 5-8. "Current knowledge of hyporheic fluxes and their spatio-temporal variability in submerged bedforms has been obtained from simulations (Boano et al., 2014; Irvine et al., 2014; Trauth et al., 2014; Stonedahl et al., 2013; Janssen et al., 2012; Cardenas and Wilson, 2007; Elliott and Brooks, 1997), laboratory experiments (Fox et al., 2014; Tonina and Buffington, 2007), and field experiments (Zimmer and Lautz, 2014; Gordon et al. 2013; Lautz and Fanelli, 2008)."

4. Is there a citation that can back up the P 12 L 6-7 statement?

Yes, there are citations to support the statement. Added in the text P12 L34 (Kunz et al., 2017; Sun et al., 2015; Gooseff et al., 2007).

The following references were added in the reference list:

Gooseff, M. N., Hall, R. O., and Tank, J. L.: Relating transient storage to channel complexity in streams of varying land use in Jackson Hole, Wyoming, Water Resources Research, 43, 2007.

Sun, N., Yearsley, J., Voisin, N., & Lettenmaier, D. P.: A spatially distributed model for the assessment of land use impacts on stream temperature in small urban watersheds, Hydrological Processes, *29*(10), 2331-2345, 2015.

Kunz, J. V., Annable, M. D., Rao, S., Rode, M., & Borchardt, D.: Hyporheic passive flux meters reveal inverse vertical zonation and high seasonality of nitrogen processing in an anthropogenically modified stream (Holtemme, Germany), Water Resources Research, 53(12), 10155-10172, 2017.

5. Section 7.2: It should be noted that dams (and many of the anthropogenic changes) can occur on small (Fanelli and Lautz, 2008) and much larger sized (Fritz and Arntzen, 2007) rivers. It would be interesting to explore the relative control of dams on such different sized systems, relative to channel slope, etc.

Yes, we agree. This is an interesting point to develop and we have added the following paragraph:

P16, L13: after "…occur."
"As a consequence, the effects of dams on HEF vary with channel planform and streambed topography. For example, in river systems characterized by large alluvial channels and unconfined aquifers, the relationship between dam-induced changes in river stage and HEF is characterized by hysteresis (Fritz and Arntzen, 2007). Therefore, HEF is not only dependent on changes in river stage but also on the difference between river and aquifer elevations (Vogt et al., 2010). As river stage varies, there is a fast response of hyporheic flows which rapidly change with the head difference within the HZ, and a slower response of HEF with changes in elevation head of the near aquifer (Fritz and Arntzen, 2007). Additionally, the lower hydraulic conductivity near the surface of the HZ, caused by accumulation of sediment in the alluvial matrix and often characterizing alluvial channels (Section 5.2), might restrict the changes in hydraulic pressure over the first cm of river sediment (Fritz and Arntzen, 2007).
In river systems characterized by small channel sizes and complex streambed morphology, differences of HEF within the subsurface upstream and downstream of dams have been attributed to the overall hydraulic behaviour around the dam and to the changes in topography induced by the dam (Hester et al., 2009; Fanelli and Lautz, 2008).
Studies using thermal sensors have reported that upstream and downstream pools created by ponding and channel degradation, respectively, have the potential to drive bedform-scale exchange flow. Temperature results suggest that the highest hyporheic exchange rates occur downstream of dams, while HEF is limited in upstream pools where fine sediment deposits yield low hydraulic conductivities (Fanelli and Lautz, 2008)."

P 2 L 2: remove first "and"
Yes, "and" removed (P2 L3)

P 2 L 3: Replace "HEF but less" to "HEF, but are less"
Yes, added "," after "HEF" (P2 L2)

P 2 L 18: Add "the" before "catchment scale"
Yes, added "the" (P2 L23)

P 3 L 25 – Change reach scale to "the reach scale" or "reach scales". There are several instances like this throughout the manuscript, please address them all.
Yes, changed "the reach scale" and accordingly throughout the manuscript. (P4 L 1)

P 3 L 28: change "HZ" to "the HZ".
Yes, changed "HZ" to "the HZ". (P4 L6)

P4 L 21 – Missing a word after "at a larger"
Yes, added "at a larger scale (i.e. catchment)". (P4 L30)

P 5 L 21 – change "though" to "through"
Yes, changed "though" with "through". (P6 L4)

P 7 L 21: change "his study" to "this study" or "their study".
Yes, changed "his study" with "their study". (P8 L3)

**Other identified changes**

Affiliations:
- "Cranfield Water Science Institute" changed into "School of Water, Energy and Environment"
- "UK" changed into "United Kingdom"

Names:
- added "C." to Robert Grabowski

Abstract:
p1 L6: changed "large and reach scale " into "at reach-scale and larger"
p1 L7: changed "and" into "to"
p1 L8: added "the"

Headers:
changed headers in Section 4: "HEF generation by in-channel bedforms" and "In-channel bedform sequences" are subheadings of "In-channel bedforms";
"Valley confinement" and "channel planform" are subheadings of "Alteration of in-channel bedform induced HEF by valley hydrology"
Subheading 4.2: corrected "an in-channel bedforms", with "in-channel bedforms"

Acknowledgements:
Added "We thank the British Geological Survey as data provider. We also thank the two anonymous reviewers for their helpful comments on the manuscript, and the colleagues that commented it in the discussion forum."

Figure 1, legend : removed "The symbol * refers to irrigation system of the adjacent agricultural fields."

Figure 2, legend: "viceversa" in italic. Added "the" to channel, reach and catchment-scale.
Table 1: formatted
Table 2: formatted

Removed excess of references in the text:
P1, L15: removed Boano et al., 2014 from references, kept only thoses highlighting ecological functioning of HZ in rivers.
P1, L18-19: removed (Boano et al., 2006, Wörman et al., 2006, Gooseff et al., 2007, Lautz et al., 2010, Cardenas and Wilson, 2007, Wondzell, 2006).
P1, L21: removed (Ward, 2016; Merrill and Tonjes, 2014)
P1, L23: deleted "sum of" and "sum of"

P2, L2-3: removed (Cardenas et al., 2004; Packman and Brooks, 2001; Elliott and Brooks, 1997), added Harvey et al., 2015.
P2, L1: changed "Turbolence" in "turbulence".
P2, L15: removed "HEF" added "affecting HEF"
P2, L17: added "sufficiently"

P2, L27: removed ".with respect to two primary topics". Added ": drivers of HEF and process interactions"

P2, L 27: changed "With respect to " into "For"

P3, L 4: removed "will"
P3, L 12: added "over"
P3, L 13: removed "a", added "spatial".
P3, L13: changed "and takes long times to return", into "with considerably longer timescale for flows to return".
P3, L18: "Consequently, this review considers large-scale GSE in addition to HEF. "moved to L14
P3, L 15: removed Sawyer et al., 2009 from references
P3, L19: added "Additionally,", removed "also"
P3, L20: removed "Finally", inserted ", and "
P3, L 22: added "Finally, "
P3, L27: added "water"to "surface", added "the"

P4, L2: added "the"
P4, L4-5: removed Wroblicky et al., 1998, Harvey and Bencala, 1993, Cardenas and Wilson, 2007,
P4, L6: removed Poole et al., 2006
P4, L14: changed "the HZ was found" with "the HZ has been found"
P4, L20: inverted "affect significantly" with "significantly affect"-
P4, L22: ((Dudley-Southern and Binley, 2015; Zimmer and Lautz, 2014); Fig. 4 in (Schmadel et al., 2017)). changed to (Dudley-Southern and Binley, 2015; Zimmer and Lautz, 2014) (Fig. 4 in Schmadel et al.(2017))

P5, L11: added "the"
P5, L14: inverted "have usually" with "usually have"
P5, L 14: removed Cardenas, 2008
P5, L22: added "the"
P5, L 23: removed "will"
P5, L 23: changed  "Sections 4.2 and 4.3" with "Section 4.1"
P5, L24: removed "(valley confinement: Section 4.5)"

P6, L6: "Hyporheic flow structure is controlled by spatial relationship of bedforms to high-
and low-permeability regions of the streambed (Stonedahl et al., 2018; Pryshlak et al., 2015;
Sawyer and Cardenas, 2009; Packman et al., 2004; Salehin et al., 2004)." moved in line L6
P6, L13: changed "into" with "in"
P6, L28: added "individual"
P6, L28, 29: removed "on its own" and "are expected to"
P6, L 33: removed Storey et al., 2003
P6, L32: added "the" before "longitudinal"

P7, L9: added "the" before "riffle-pool"

P9, L10. Changed "pressure" to "head"
P9, L26: changed "clues"  into "evidences"

P10, L 12: removed "forefront" with "priority"
P10, L 13: added "the" before "stream"
P10, L17: inserted "should", changed "provide" to "generate"
P10, L 18: added "a" before "result"

P10, L27-28: striked out "Given the direct coupling of stream and pore water flow," and "generally greatly". Added "Hyporheic"

P11, L 4: removed "and" before "shallower"
P11, L5 : added "Furthermore"
P11, L6: change "having" into "with". Refrased "despite that in coarser bed material, fine sediments accumulate and clog pores" into ", but connectivity can be reduced by the accumulation of fine sediment that clogs pores"
P11, L17: changed "result in fact from" in "are caused by"
P11, L21: striked out "can"
P11, L 23: Added "in these river types"

P12, L12: changed "appear" to "exist"
P12, L13: changed "structure" to "bedrock"
P12, L13: removed "including limestone, dolomite, shale, siltstone, sandstone, and conglomerate "
P12, L14: changed "e.g." into "i.e."
P12, L15: "surface-subsurface exchange" changed to "GSE"
P12, L 20: "aquitard" changed in "geological strata" and added "(i.e. aquitards)"
P12, L 21: "removed If the confinement is due to the presence of near-surface bedrock,"

P13, L4: removed "Certainly" and capitalized "In-channel". Removed "the" in front of "create"
P13 L5: removed (Section 6.1), changed "two ecological functions" to "three key types"
P13, L10, 11: added "velocities"
P13, L22: removed "mixing due to vegetation and to"
P13, L22: added "the"
P13, L26: changed "wood deposits" into "accumulations of wood"

P14, L23: added "and HEF"
P14, L 30: "mixing" changed to "the flow of water through sediment into the root comparable to"
P14, L20 : "daily" changed to "diurnal"
P14, L33: added "effects on"

P15, L3: changed "energy" in "water"
P15, L30: added "the"

P16, L6: Moved "For example" at the beginning of the sentence

P17, L17: "First, we review" instead of "We first review"
P17, L18: removed "that" that is typed twice
P17, L25: moved "streambed" and added "geological"
P17, L26: replaced "Because of" with "Given"

[revised manuscript text omitted]